# Biodiversity across trophic levels drives multifunctionality in highly diverse forests

Andreas Schuldt [1,2], Thorsten Assmann [3], Matteo Brezzi[4,5], François Buscot[1,6], David Eichenberg[1,7], Jessica Gutknecht[6,8], Werner Härdtle[3], Jin-Sheng He[9], Alexandra-Maria Klein[10], Peter Kühn [11], Xiaojuan Liu[12], Keping Ma[12], Pascal A. Niklaus[4], Katherina A. Pietsch [7], Witoon Purahong [6], Michael Scherer-Lorenzen [13], Bernhard Schmid [4], Thomas Scholten [11], Michael Staab [10,14], Zhiyao Tang[9], Stefan Trogisch [1,2,13], Goddert von Oheimb [1,15], Christian Wirth[1,7], Tesfaye Wubet [1,6,16], Chao-Dong Zhu[17] & Helge Bruelheide [1,2]

Human-induced biodiversity change impairs ecosystem functions crucial to human well-being. However, the consequences of this change for ecosystem multifunctionality are poorly understood beyond effects of plant species loss, particularly in regions with high biodiversity across trophic levels. Here we adopt a multitrophic perspective to analyze how biodiversity affects multifunctionality in biodiverse subtropical forests. We consider 22 independent measurements of nine ecosystem functions central to energy and nutrient flow across trophic levels. We find that individual functions and multifunctionality are more strongly affected by the diversity of heterotrophs promoting decomposition and nutrient cycling, and by plant functional-trait diversity and composition, than by tree species richness. Moreover, cascading effects of higher trophic-level diversity on functions originating from lower trophic-level processes highlight that multitrophic biodiversity is key to understanding drivers of multifunctionality. A broader perspective on biodiversity-multifunctionality relationships is crucial for sustainable ecosystem management in light of non-random species loss and intensified biotic disturbances under future environmental change.

[1] German Centre for Integrative Biodiversity Research, (iDiv) Halle-Jena-Leipzigv, Deutscher Platz 5e, 04103 Leipzig, Germany. [2] Institute of Biology/Geobotany and Botanical Garden, Martin-Luther-University Halle-Wittenberg, Am Kirchtor 1, 06108 Halle (Saale), Germany. [3] Institute of Ecology, Leüphana University Lüneburg, Scharnhorststrasse 1, 21335 Lüneburg, Germany. [4] Department of Evolutionary Biology and Environmental Studies, University of Zurich, Winterthurerstrasse 190, 8057 Zurich, Switzerland. [5] Institute of Global Health, University of Geneva, 9 Chemin des Mines, 1202 Geneva, Switzerland. [6] Department of Soil Ecology, UFZ-Hemholtz Centre for Environmental Research, Theodor-Lieser-Strasse 4, D-06120 Halle (Saale), Germany. [7] Department of Systematic Botany and Biodiversity, Leipzig University, Johannisallee 21-23, 04103 Leipzig, Germany. [8] Department of Soil, Water, and Climate, University of Minnesota, Twin Cities, 1991 Upper Buford Circle, St. Paul, MN 55108, USA. [9] Key Laboratory for Earth Surface Processes of the Ministry of Education, Department of Ecology, College of Urban and Environmental Sciences, Peking University, 100871 Beijing, China. [10] Nature Conservation and Landscape Ecology, University of Freiburg, Tennenbacher Strasse 4, 79106 Freiburg, Germany. [11] Chair of Soil Science and Geomorphology, Eberhard Karls-University of Tübingen, Rümelinstrasse 19-23, 72720 Tübingen, Germany. [12] Institure of Botany, Chinese Academy of Sciences, Beijing 100093, China. [13] Faculty of Biology, Geobotany, University of Freiburg, Schaenzlestrasse 1, 79104 Freiburg, Germany. [14] Freiburg Institute of Advanced Studies (FRIAS), University of Freiburg, Albertstrasse 19, 79104 Freiburg, Germany. [15] Institute of General Ecology and Environmental Protection, Technische Universität Dresden, Pienner Strasse 7, D-01737 Tharandt, Germany. [16] Department of Community Ecology, UFZ-Helmholtz Centre for Environmental Research, Theodor-Lieser-Strasse 4, D-06120 Halle (Saale), Germany. [17] Key Laboratory of Zoological Systematics and Evolution, Institute of Zoology, Chinese Academy of Sciences, Beijing 100101, China. Deceased: Matteo Brezzi  Correspondence and requests for materials should be addressed to A.S. (email: andreas.schuldt@idiv.de)

Concerns over detrimental effects of human-induced biodiversity loss on ecosystem functions have spawned extensive research on biodiversity–ecosystem functioning (BEF) relationships[1]. Over the last few decades, research has expanded from the study of small-scale and grassland systems, and individual ecosystem functions to larger-scale analyses of ecosystems—such as forests—and multifunctionality[2–4]. Multifunctionality describes the fact that ecosystems simultaneously provide—and are often valued and managed for—a multitude of ecosystem functions essential to human well-being[5]. Recent studies have highlighted the great potential of species-rich forests for promoting and securing multifunctionality[6–10]. These findings are important for the development of sustainable forest management strategies that can take into account a multitude of stakeholder requirements[11]. However, significant knowledge gaps remain with respect to the relative importance of different facets of biodiversity across trophic levels for multifunctionality and the transferability of results to more biodiverse systems than studied so far[12].

In particular, we require a thorough integration of higher trophic levels into analyses of multifunctionality, as they substantially modify key ecosystem functions, such as primary productivity and nutrient cycling[13,14]. However, there is still a poor understanding of—and an insufficient distinction between—higher trophic-level functions as potentially important components of multifunctionality and the biodiversity of higher trophic levels as direct predictors of biodiversity–multifunctionality relationships. With respect to the former, ecosystem functions such as herbivory and microbial activity have previously been included as components of multifunctionality in forest studies[7–10]. However, other higher trophic-level functions have received less attention, despite their potentially important contribution to multifunctionality. For example, herbivory is regulated by predation and parasitism[15], but these two functions have rarely been included directly in multifunctionality analyses. And yet, such biocontrol functions may be a critical aspect of forest management in a changing climate, where the frequency and intensity of biotic disturbances, such as insect herbivore outbreaks, will likely increase[16].

Perhaps even more important, however, is that the diversity of higher trophic-level organisms may serve as a key predictor of multifunctionality[17] and therefore as a management target. Studies on multifunctionality sometimes consider heterotrophic diversity as one of the many components of multifunctionality[10,18] rather than as a direct driver of ecosystem functioning. However, this neglects the important influence of higher trophic-level diversity on many ecosystem functions[17,19] and hinders a full understanding of overall biodiversity effects on multifunctionality. Plant diversity is not necessarily a suitable surrogate for heterotrophic diversity[20] and accounting for the latter may improve models on biodiversity–multifunctionality relationships[19]. Nevertheless, our understanding of the interconnections between multiple ecosystem functions and of the extent to which the impact of higher trophic-level diversity on multifunctionality propagates through food webs is poor for most ecosystems. A multitrophic perspective might be particularly important for low-latitude forests, where trophic interactions of highly diverse communities influence ecosystem structure and functioning[21,22].

For these low-latitude forests in general, we require more information on how changes in biodiversity affect multifunctionality. Subtropical and tropical forests are characterized by a high biodiversity and a global importance for the provisioning of key ecosystem services[23]. However, most studies exploring biodiversity-multifunctionality relationships are based on data from temperate and boreal forests (but see, e.g., ref. [24]). A recent study[8] highlighted that BEF relationships for many functions vary substantially with changing environmental conditions, implying that findings from temperate and relatively species-poor regions are not necessarily transferrable to species-rich subtropical and tropical regions. This may be particularly the case when considering the likely stronger influence of biotic interactions across trophic levels for the latter[22]. It is also not clear whether the strong effects of plant species richness on multifunctionality observed in previous, species-poor forests[6,8–10] are transferable to species-rich forests. Previous studies have often reported a leveling-off of such effects at higher levels of plant species richness[2]. In such cases, metrics of functional diversity and community composition might be more meaningful predictors than plant species richness alone[24,25]. Understanding the relevance of such additional biodiversity predictors may also be important to predict biodiversity effects under changing environmental conditions, because community-level drivers of biodiversity change may often have stronger effects on species composition than on species richness[26].

Here we adopt a multitrophic perspective to analyze the relative influence of multiple facets of biodiversity on multifunctionality for the first time in a highly diverse subtropical forest system. We do this by incorporating the species richness of invertebrate and microbial functional groups, woody plant functional diversity and composition, and woody plant species richness. Our analysis of multifunctionality considers 22 independent measurements of nine important ecosystem functions that reflect the ecological impact of key interactions within and across trophic levels on energy and nutrient flow, and which can be related to important ecosystem services (erosion control, microbial activity, primary productivity, nitrogen cycling, leaf decomposition, wood decomposition, herbivory resistance, predation, and parasitism; Supplementary Table 1). We quantified multifunctionality for 27 forest stands with two commonly used approaches, the average multifunctionality and the multiple threshold approach[5]. We related the multifunctionality indices to the species richness of important functional groups of animals and microorganisms, and to species richness, functional diversity, trait, and species composition of the woody plant communities, while controlling for environmental conditions (see Methods). In addition, we tested the extent to which effects of higher trophic-level diversity on multifunctionality propagate through the food web by analyzing the contribution of the species richness of individual trophic levels to multifunctionality after excluding functions directly modified by a given trophic level and by testing for effects of overall community richness across trophic levels.

For such biodiverse forests, we hypothesize that multifunctionality is strongly affected by the species richness of higher trophic levels—not only when taking into account a wider spectrum of ecosystem functions mediated by interactions across trophic levels but also when excluding functions directly mediated by a given trophic level (i.e., considering only functions primarily mediated by adjacent trophic levels). Moreover, we expect that the functional composition and diversity of plant communities contribute significantly to explaining multifunctionality beyond the effects of plant species richness alone. Our study shows that diversity effects of individual trophic levels on ecosystem multifunctionality cascade through the food web, highlighting the need for a multitrophic perspective when trying to disentangle the drivers of BEF relationships.

## Results

**Overall drivers of ecosystem multifunctionality.** Higher trophic-level species richness significantly influenced both average and threshold-based multifunctionality. Model averaging revealed

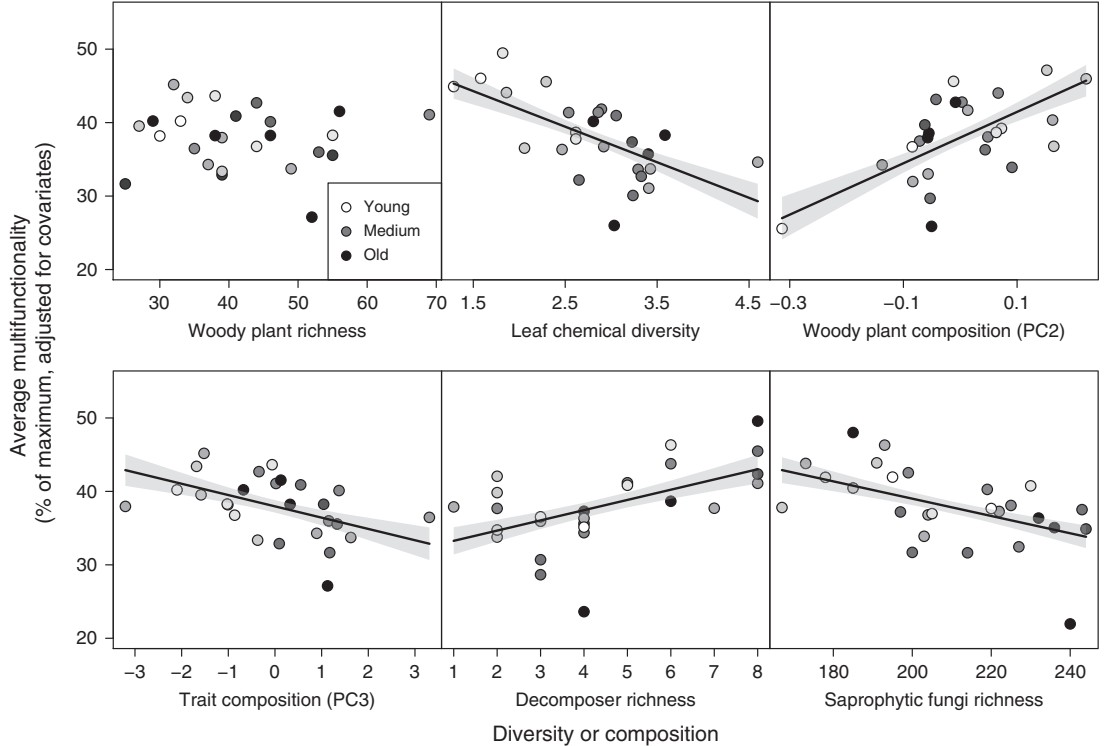

**Fig. 1** Biodiversity effects on average multifunctionality. Multimodel-averaging results for effects of woody plant diversity and composition, and heterotrophic species richness on multifunctionality as the average of nine standardized ecosystem functions in a biodiverse subtropical forest. Only variables retained after model simplification and model averaging (as well as tree species richness for comparison) are shown. Values on the x-axis represent either increasing diversity or differences among study plots in species or functional-trait composition. Note that y-axis values show data adjusted for covariates (see Supplementary Fig. 11 for raw data). Regression lines ( ± 1 SE, fitted across all 26 plots) are adjusted for covariates and indicate significant ($P \leq 0.05$) relationships. The stand age of the study plots is indicated by a continuous gradient from white (youngest plots ~20-year-old) to black (oldest plots > 80-year-old)

particularly strong positive effects of the species richness of macrofaunal decomposers and negative effects of the richness of saprophytic fungi (Fig. 1, Supplementary Figs 1–4, Supplementary Tables 2, 3). Multifunctionality was also particularly strongly related to the plant-based predictors of leaf chemical diversity, which negatively affected multifunctionality, and to woody plant species composition (positive effects of PC2; Supplementary Table 4) (Fig. 1, Supplementary Figs 1–4, Supplementary Tables 2 and 3). Path analyses confirmed the direct effects of these predictors on multifunctionality (Fig. 2, Supplementary Table 5). In addition, it revealed indirect effects of woody plant species and functional-trait composition on multifunctionality mediated by decomposer species richness. In contrast, the effect of the species richness of saprophytic fungi on multifunctionality was not significantly influenced by woody plant community metrics, and there was no significant linkage between the effects of the richness of decomposers and of saprophytic fungi (Fig. 2).

Average multifunctionality per plot ranged from 20% to 50% of the observed maximum multifunctionality. The predictors accounted on average for 66% of the variability in multifunctionality among plots and were predicted to cause changes in multifunctionality between 23% (species richness of saprophytic fungi) and 102% (woody plant composition PC2) (Supplementary Table 2, Supplementary Fig. 1). In addition, there was a significant effect of trait composition (negative effect of PC3, which was negatively related to leaf dry matter content (LDMC), leaf carbon (C) content, and xylem vessel diameter; Supplementary Table 6) on average multifunctionality (Fig. 1, Supplementary Table 2).

When calculating multifunctionality based on threshold models, they explained on average 67% (30% threshold), 54%

(60% threshold), and 49% (90% threshold) of the data variability. The average number of the nine functions larger than the threshold decreased from 6.9 (30%) to 3.1 (60%) and 0.9 (90%) (Supplementary Fig. 2). Individual predictors increased or decreased the number of functions exceeding a given threshold by up to 72% (from 5.9 to 8.2 functions, predictor: parasitoid species richness) at the 30% threshold, and up to 112% (from 1.9 to ~0 functions, predictor: leaf chemical diversity) at the 90% threshold (Supplementary Table 3, Supplementary Figs 2–4). The negative effects of leaf chemical diversity and trait composition (PC3) were consistent across the three thresholds (Supplementary Table 3, although being nonsignificant for both at the 30% threshold and for trait composition at the 90% threshold). Woody plant composition (PC2) and macrofaunal decomposer richness had significant positive effects on multifunctionality at two of the three thresholds, whereas the species richness of saprophytic fungi significantly affected multifunctionality only at the highest threshold (90%, with a nonsignificant effect at the 30% threshold) (Fig. 1). Additional significant positive effects of predator and parasitoid species richness were only apparent at one of the thresholds (Supplementary Table 3, Supplementary Fig. 2).

## Linkages across trophic levels and multidiversity effects. We also found significant diversity effects on multifunctionality when analyzing these relationships on individual trophic levels in separate analyses, excluding functions directly mediated by the given trophic level being analyzed (e.g., parasitism for parasitoids, predation for predators). When considered separately, four of

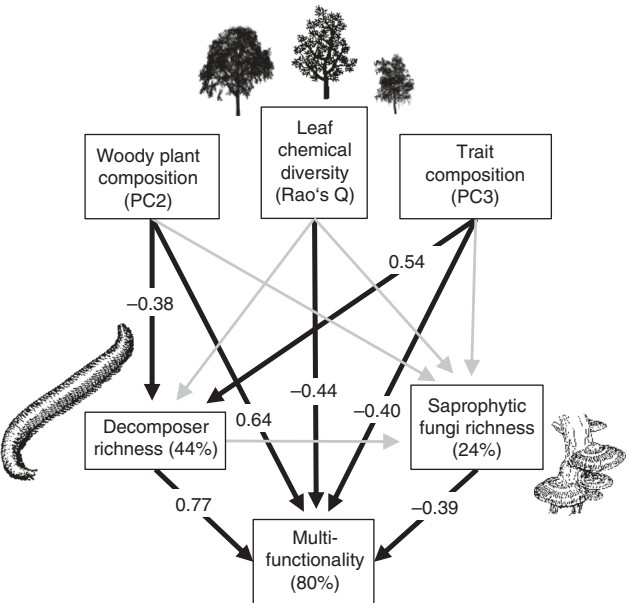

**Fig. 2** Path model of biodiversity effects on average multifunctionality. Combined effects and interdependencies of the diversity and composition of woody plant communities and heterotrophic species richness on multifunctionality ($\chi^2 = 1.9$, DF = 3, $P = 0.598$). Black arrows (with standardized path coefficients) indicate significant ($P \leq 0.05$) effects, grey arrows show nonsignificant effects. Percentage values are explained variance of endogenous variables. The plant, fungi, and animal icons (from www.openclipart.org) are licensed for use in the public domain without copyright (Creative Commons Zero 1.0)

the six trophic levels showed significant diversity effects on multifunctionality that were not dependent on functions directly mediated by the respective trophic level. This indicates that these diversity-multifunctionality relationships propagate through the food web (Fig. 3, Supplementary Table 7). This was also the case for the species richness of parasitoids and herbivores, whose effects on multifunctionality were masked by stronger diversity predictors in the overall analyses when the diversity of all functional groups was considered simultaneously. The effects of decomposer species richness on multifunctionality after excluding leaf and wood decomposition depended on stand age and were only marginally significant (Fig. 3, Supplementary Table 7). As these comparisons are interdependent, significance levels need to be interpreted with care. Nevertheless, significant effects of species richness in four of the models (with $\alpha = 0.05$ we would expect only one out of the six comparisons to show a significant effect by chance) and the results of the overall model above support the biological relevance of our results. In general, the multifunctionality indices excluding individual functions were highly correlated with overall multifunctionality based on all functions (Supplementary Table 8), showing that none of the individual functions disproportionately influenced overall multifunctionality (Supplementary Figs 5, 6).

Although the main effects of average total community richness, calculated as the average of the standardized species richness indices across all trophic levels, and stand age were not significant (Supplementary Fig. 7b), total community richness positively affected ecosystem multifunctionality in older forest stands, but not in young forest stands (Fig. 4, Supplementary Table 9). Excluding tree species richness and focusing only on heterotrophic community richness yielded congruent results (Supplementary Fig. 7c, Supplementary Table 9), because both community richness metrics were highly correlated (Pearson's $r = 0.98$, $P < 0.001$).

**Drivers of individual ecosystem functions**. The significant predictors of average and threshold-based multifunctionality also had significant effects on at least one (and up to four) of the nine ecosystem functions when functions were analyzed individually (Fig. 5, Supplementary Figs 8, 9, Supplementary Table 10). The direction of these effects was largely consistent with the direction of the effect for the strongest predictors in the multifunctionality analyses. Soil-associated ecosystem functions (e.g., erosion control, microbial activity) were strongly related to environmental plot conditions and the species richness of belowground organisms. Aboveground functions mediated by higher tropic levels (herbivory resistance, predation, parasitism) were more strongly affected by aboveground heterotrophic species richness and plant diversity or composition (Fig. 5, Supplementary Figs 8, 9, Supplementary Table 10).

## Discussion

Our analysis of 22 independent measures of nine ecosystem functions in a biodiverse subtropical forest revealed strong effects of the species richness of heterotrophic organisms, as well as of plant functional-trait diversity and composition, on both individual functions and multifunctionality. Importantly, the effects of higher trophic-level species richness propagated through the food web, indicating that higher trophic levels have important indirect effects on ecosystem functions driven by adjacent trophic levels. Including multiple facets of biodiversity and composition in addition to woody plant species richness helped to explain up to 67% of the variation in multifunctionality among study plots, showing that these complementary components of biodiversity may be important drivers of ecosystem multifunctionality. Previous studies have mostly highlighted the effects of tree species richness on multifunctionality in forests, but have been conducted in comparatively species-poor regions, such as temperate and boreal Europe[6,8–10]. The extent to which functional diversity of plants and animals influence multifunctionality in these forests remains to be tested, although first results indicate an important role of species composition in these forests as well[8]. Changes in overall biodiversity and species composition beyond species loss may therefore be key to understanding and managing multifunctionality not only in highly diverse ecosystems.

The species richness of several groups of heterotrophic organisms contributed significantly to explaining changes in multifunctionality, both when considering overall multifunctionality and when excluding functions directly driven by a given trophic level. Moreover, overall multidiversity across trophic levels was positively related (except in young successional forest stands) to multifunctionality. This sheds new light on the growing evidence from other ecosystems that higher trophic-level diversity is crucial to understanding the effects of biodiversity on ecosystem multifunctionality[7,17,19,27]. Our results emphasize that a focus on the producer level might underestimate the overall effects of biodiversity on ecosystem functions and multifunctionality. Thus, paying closer attention to higher trophic-level diversity (and composition[28]) may help to move forward ecological theory and application[19]. Organisms involved in the decomposition of organic material and nutrient cycling had a particularly important role at our study site, probably because the activity of these groups directly or indirectly connects many ecosystem functions. Decomposers promote leaf and wood decomposition, which influence erosion control[29] and strongly affect nutrient availability for plant growth and higher trophic levels[30]. Macrofaunal decomposer species richness was positively related to multifunctionality, in accordance with the assumption that more species-rich decomposer communities contribute more effectively to decomposition processes[31]. Effects of decomposer

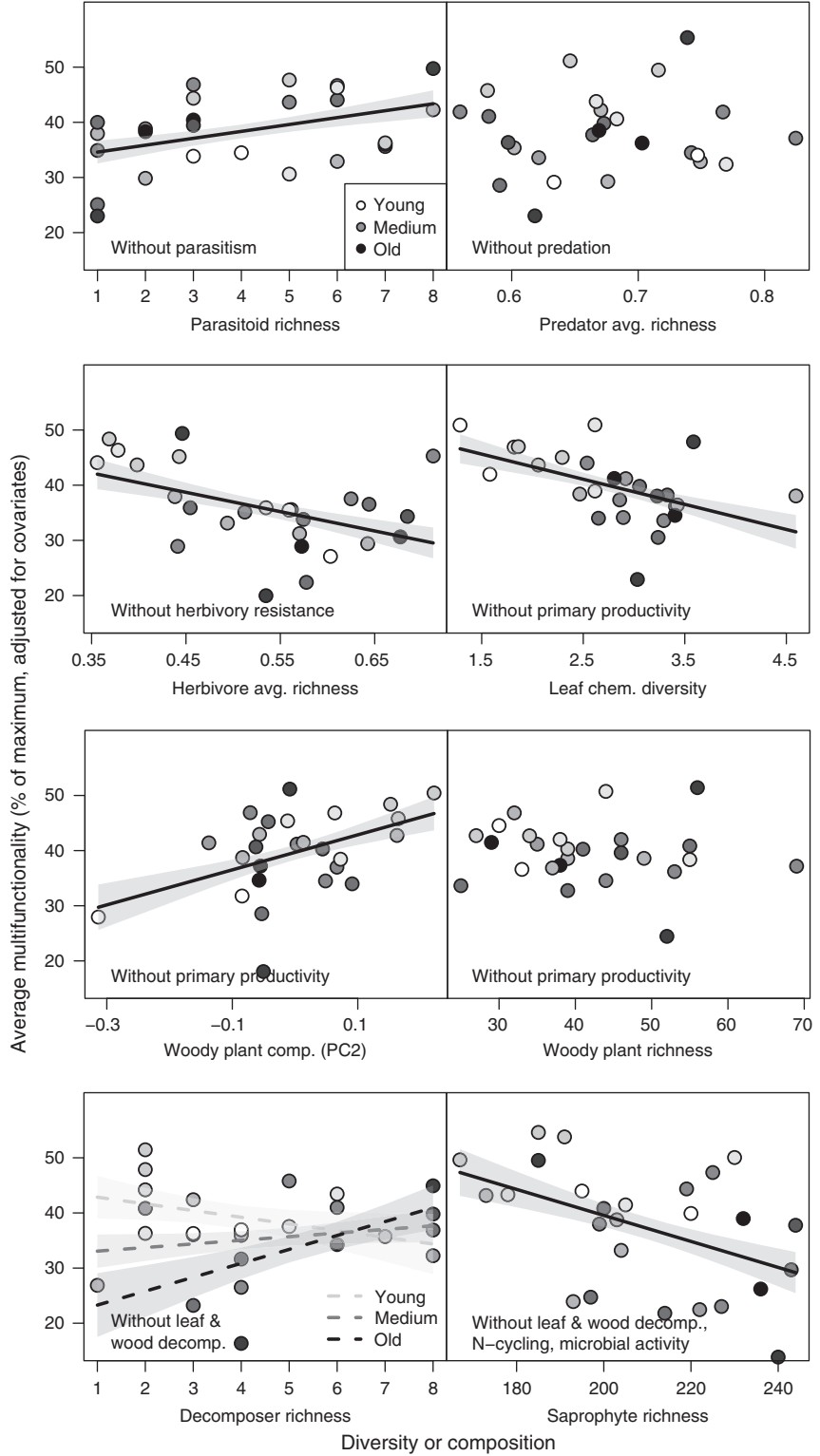

**Fig. 3** Biodiversity effects within individual trophic levels on average multifunctionality. Effects of individual models on woody plant diversity and composition, or heterotrophic species richness of individual trophic levels on multifunctionality as the average of five to eight standardized ecosystem functions (excluding functions directly mediated by a given trophic level). Values on the *x*-axis represent either increasing diversity or differences among study plots in species composition. Note that *y*-axis values show data adjusted for covariates (see Supplementary Fig. 12 for raw data). Solid regression lines ( ±1 SE, fitted across all 26 plots, except for decomposer diversity, where lines are model predictions for young (40 years), medium (70 years), and old (100 years) forest stands) are adjusted for covariates and indicate significant ($P \leq 0.05$) relationships. Broken lines indicate marginally significant ($P < 0.07$) relationships. The stand age of the study plots is indicated by a continuous gradient from white (youngest plots ~20-year-old) to black (oldest plots > 80-year-old); avg. average

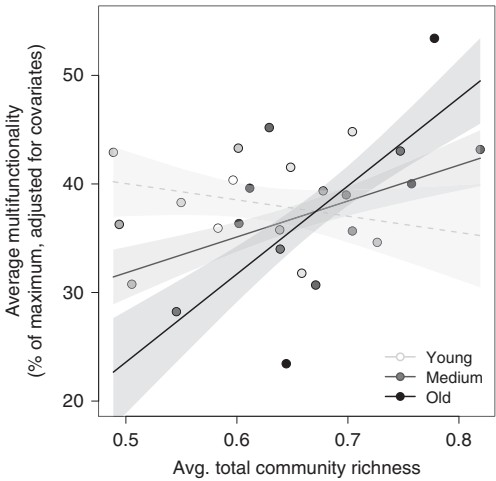

**Fig. 4** Overall multidiversity effects on average multifunctionality. Effects of average total community richness (average of standardized species richness across trophic levels, including tree species richness) on multifunctionality as the average of nine standardized ecosystem functions. Solid regression lines (model predictions for young (40 years), medium (70 years), and old (100 years) forest stands) indicate significant ($P \leq 0.05$) relationships. The broken line indicates nonsignificant relationships. Note that y-axis values show data adjusted for covariates (see Supplementary Fig. 7a for raw data). Regression lines (± 1 SE, fitted across all 26 plots) are adjusted for covariates and indicate significant ($P \leq 0.05$) relationships. The stand age of the study plots ($n = 26$) is indicated by a continuous gradient from white (youngest plots ~20-year-old) to black (oldest plots > 80-year-old); avg. average

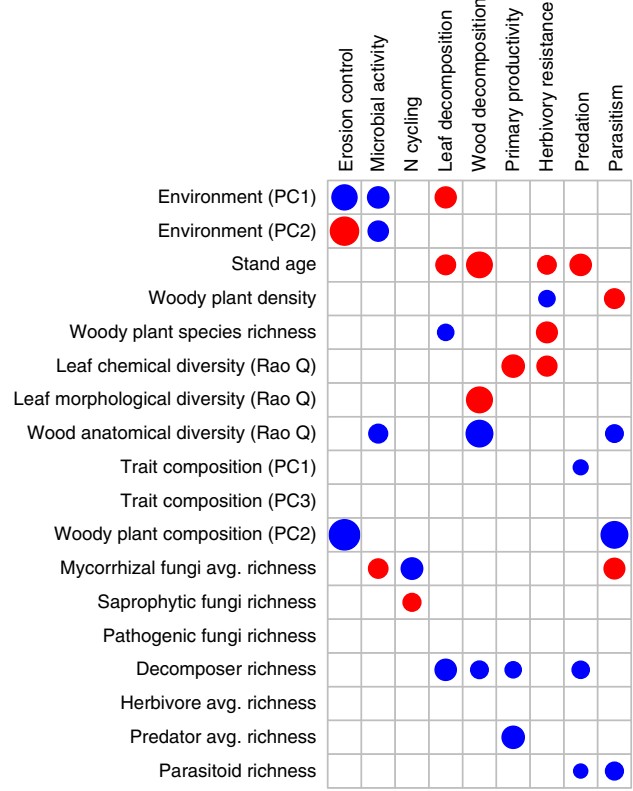

**Fig. 5** Biodiversity effects on individual functions. Summary of the model-averaged effects (with a maximum of four predictors per individual model) of abiotic plot characteristics, woody plant diversity and composition, and heterotrophic species richness on nine ecosystem functions in a biodiverse subtropical forest. Blue and red circles indicate significant ($P \leq 0.05$) positive and negative effects, respectively, in the models shown in Supplementary Table 10 ($n = 26$) (see also Supplementary Figs 8, 9). Circle size scales with the predictors' standardized estimate. avg. average, PC principal component

richness were most obvious when multifunctionality relationships were controlled for the influence of the diversity of other taxa. This was confirmed by path analyses, which further showed that indirect effects of woody plant composition on multifunctionality via decomposer species richness weakened direct plant effects on multifunctionality. Interestingly, the species richness of sapro-phytic fungi showed contrasting, negative effects on multi-functionality. The same was the case for herbivore average species richness in the trophic level-specific analysis, although potentially negative effects on individual functions—such as primary pro-ductivity, herbivore resistance, or decomposition—were com-paratively weak and masked by other, stronger diversity predictors in our analyses. A high species richness of saprophytic fungi might reduce the dominance, or influence the performance, of individual and potentially highly effective fungal species in decomposition[32]. This could lead to interactions between co-occurring micro-organisms or competition between plants and fungi that affect soil-available nutrients, and potentially negatively affect primary productivity and associated ecosystem functions[33,34]. In highly diverse communities, saprophytic fungi have been shown to invest more energy and resources into antagonistic interactions (i.e., production of secondary metabolites) than into growth and the production of extracellular enzymes for decomposition[35,36]. The biosynthesis of such extracellular enzymes is highly nutrient demanding and energetically expensive for fungi, as these bioca-talysts have to be secreted in substantial amounts, which means substantial losses of C and N. These trade-offs might explain the low enzyme activity and decomposition rates in forest stands with a high species richness of saprophytic fungi[35,36].

The important role of higher trophic-level species richness in complementing plant-based effects on multifunctionality also becomes evident when considering ecosystem functions that have not been included in previous biodiversity–multifunctionality studies, such as predation, parasitism, and erosion control. These

functions are linked to key ecosystem services (soil fertility and biocontrol) and are—particularly in the case of predation and parasitism—dependent on the performance of higher trophic levels less directly affected by changes in plant diversity than lower trophic levels and their functions[37]. This expectation is corroborated by previous studies at our study site on the species richness and abundance of individual groups of predators, which revealed only limited evidence for positive effects of woody plant species richness on these groups[38–40]. Moreover, resistance against insect herbivory was negatively related to woody plant species richness (Fig. 4; ref. [41]). This demonstrates that even functions found to benefit from increased tree species richness in some forests[42] may show deviating and even opposite relation-ships in other forest systems, which emphasizes the context-dependency of BEF and multifunctionality studies[8,43]. Including a wider set of ecosystem functions is important for sustainable forest management particularly in the face of climate change-induced management challenges, such as increased frequency and intensity of insect herbivore outbreaks[16]. In this context, the strong, positive effect of predator average species richness on primary productivity observed in our study (see Supplementary Table 10)—indicative of increasing top-down control of herbi-vores—is highly relevant. Strengthening resistance against her-bivores and biocontrol by predators and parasitoids may contribute to developing suitable strategies that ensure a stable provisioning of ecosystem services under uncertain future

environmental conditions. As these functions are not necessarily strongly and directly mediated by plant species richness, trade-offs with other functions may be stronger than what has been observed among functions in previous multifunctionality analyses[10]. Such potential trade-offs could complicate attempts to promote multifunctionality by means of increasing plant species richness[5].

In addition to heterotrophic species richness, woody plant species composition, trait composition, and the functional diversity of chemical leaf traits were strong predictors of ecosystem multifunctionality. Leaf traits may be important for many ecosystem functions because these traits determine key aspects of the life history strategies of plants[44]. Interestingly, the effects of leaf chemical diversity were negative, and multifunctionality was higher in chemically less diverse communities. Possibly, higher diversity coincides with decreased dominance of individual species and their traits. As dominant species can disproportionately drive ecosystem functions[24,45], a decrease in dominance with increasing diversity might result in overall negative effects. In the case of resistance to herbivory, which was one of the functions particularly negatively affected by leaf chemical diversity, leaf damage was probably promoted by providing more diverse diets to dominant generalist herbivores[46,47]. However, a large proportion of the variability in multifunctionality was explained by changes in woody plant species composition that were not directly related to leaf- or wood-trait composition and diversity (represented by PC2 of woody plant composition). Thus, when considering such a wide range of functions, information based on a limited set of plant traits might not be sufficient to fully capture multifunctional variability. The additional effects of woody plant composition might therefore reflect the influence either of traits not considered in our analyses, or of complex interactions among traits that our diversity and trait composition metrics did not account for. Alternatively, they might also indicate an important role of species identities. Most of the species with highest loadings on PC2 of woody plant composition were highly abundant tree or shrub species (Supplementary Table 4; ref. [48])—such as *Schima superba*, a dominant tree species in the study region. A previous study in tropical forests emphasized the important role of dominant species in driving ecosystem multifunctionality[24], and our results might indicate the influence of such species in driving functions independent of major trait compositional patterns (as represented by PC1 of woody plant composition; see Methods).

In part, the effects of trait diversity and species composition might be related to changes in successional and environmental conditions. The study plots were established along a successional gradient, and stand age showed significant (negative) effects on several of the individual functions considered in our study. Likewise, environmental conditions particularly influenced soil-related functions such as erosion control and microbial activity. Several predictors were moderately correlated with, and therefore potentially influenced by, stand age or environment (e.g., leaf chemical diversity as the variable most strongly correlated with stand age: Pearson's $r = 0.61$, $P < 0.001$). However, the fact that the biotic attributes of the plant communities were stronger predictors of multifunctionality than abiotic variables, and that stand age or diversity interactions with stand age had only limited predictive power (as they were only retained in the final model of overall multidiversity but in none of the other models), suggests that changes in biotic attributes are key to mechanistically explaining multifunctionality. Previous studies at our study site found only few tree species that were specific to individual successional stages[48], and that woody plant functional diversity was maintained at a constant level throughout succession[49], which might contribute to explaining the largely weak effects of stand age in our study. We note the observational character of our study

and that manipulative experiments are required to confirm causality and to disentangle underlying mechanisms. Nevertheless, our results are consistent with the finding of ref. [39] that community patterns at our study site revealed many linkages among key functional groups of organisms that were independent of species richness. Moreover, our findings demonstrate that such linkages across trophic levels[39] make adopting a multitrophic perspective critical for understanding ecosystem functioning[50]. The weak effects of woody plant species richness observed in our study contrast with findings of previous studies on multifunctionality in forests[6,8–10], but might be explained by the much higher woody plant species richness of our forest stands. Biodiversity effects are often particularly pronounced at lower levels of species richness, and our findings might reflect the often expected levelling-off of diversity effects at higher levels of species richness[25]. Nevertheless, some of the functions we examined did show significant relationships with woody plant species richness (leaf decomposition, herbivory resistance). This indicates that although plant species richness can influence individual ecosystem functions even at high richness levels, these effects may not be strong enough to influence overall multifunctionality in our study system.

Our study shows that under real-world conditions the species richness of key functional groups of heterotrophic organisms, as well as plant species composition and functional-trait diversity, may be decisive moderators that need to be considered in addition to plant species richness when trying to understand and utilize the drivers of multifunctionality for management. The effects of higher trophic levels propagated through the food web, showing that their contribution to multifunctionality extends far beyond effects on the functioning within individual trophic levels. This is important because future environmental conditions are predicted to lead to an intensification of biotic disturbances, which essentially are of multitrophic nature[16], and because environmental change may lead to non-random loss of species and compositional changes in many cases[51]. Integrating a broader spectrum of functional effects of higher trophic levels and their diversity into multifunctionality analyses may therefore be highly relevant for sustainable management strategies aimed at securing forest systems against future environmental changes. However, our study indicates that with an increasing range of ecosystem functions trade-offs that influence the optimization of multifunctionality may become more apparent.

## Methods

**Study site**. The study was conducted in the Gutianshan National Nature Reserve (29°140' N; 118°070' E) in Zhejiang Province, south-east China. The reserve, situated in mountainous terrain (250–1260 m above sea level), covers ca. 8000 ha of semi-evergreen, broadleaved forest. The climate is subtropical, with a mean annual temperature of 15.3 °C and mean annual precipitation of ca. 2000 mm.

In 2008, we established 27 study plots (30 m × 30 m) following a stratified random selection design based on the stand age (< 20 to > 80 years) and woody plant species richness (25–69 species) typically encountered in the reserve[48]. We excluded one of the study plots from our analyses, because extreme values indicated processing errors of the soil samples used for the measurement of several belowground ecosystem functions that would have biased the statistical analyses (see Supplementary Fig. 10).

**Ecosystem functions**. We used 22 independent measurements of nine ecosystem functions that play important roles in energy and nutrient flow across trophic levels in our study system. These nine functions were: erosion control, microbial activity, primary productivity, nitrogen cycling, leaf decomposition, wood decomposition, herbivory resistance, predation, and parasitism (Supplementary Table 1).

Erosion control was quantified as the inverse of the estimated erosivity of rain drops. Microbial activity comprised measurements of enzyme activity and microbial biomass. We measured the activity of four enzymes that play important roles in decomposition processes and nutrient cycling by degrading cellulose (β-xylosidase), xylan (xylosidase), chitin (N-acetyl-glucosaminidase), and polyphosphates (acid phosphatase). Primary productivity was estimated as the increment in stem basal area of woody plants over four years. Soil nitrogen cycling

was assessed by measuring net ammonification and net nitrification. The quantification of leaf decomposition comprised measurements of community-level and species-specific leaf decomposition rates. Wood decomposition was assessed by two complementary approaches: calculation of community wood decomposition rates and measurement of decomposition rates of standardized wood samples. Herbivory resistance was based on measurements of leaf damage by herbivorous insects of canopy trees and woody plant saplings. Predation was measured for two groups of common arthropod predators, ants and predatory wasps. Parasitism rates were assessed for parasitic Hymenoptera. See Supplementary Table 1 for full details on all measurements.

Our study provides a comprehensive analysis of these functions in a multifunctional and multitrophic context, which differs substantially in scope from previous analyses in our study system that looked at individual functions and their relationship with woody plant communities (Supplementary Table 11).

**Biotic attributes**. We conducted a complete inventory of the woody plant communities of each plot (abundance, species composition), with all tree and shrub individuals > 1 m height, in 2008. We estimated the stand age of the study plots from tree stem cores and diameter at breast height (DBH) measurements (to the closest 0.1 mm)[48]. To characterize the functional composition of the woody plant communities, we measured a range of leaf morphological, leaf chemical, and wood anatomical traits of the woody species: leaf area (LA), specific LA and LDMC, leaf C content, leaf carbon to nitrogen (C:N) ratio, leaf polyphenolics content, wood density, mean xylem vessel diameter, and wood fiber wall thickness. These traits determine key aspects of the life history strategies of plants[44] and have previously been shown to strongly affect primary productivity, decomposition, herbivory, and other important ecosystem functions in subtropical and other forests[46,52,53]. Leaf traits were measured for ca. 80% of the 147 woody plant species recorded on the study plots and these species represented 95% of the total number of tree and shrub individuals at the study sites. Samples for trait measurements were taken from sun-exposed leaves of five to seven plant individuals in total[54], collected from up to seven plots per species in the summer of 2008. Trait measurements followed standardized protocols[55]. Wood traits were available for 93 species that represented 83% of the total number of woody plants at the study site. Wood samples were taken from sun-exposed branches with a diameter of at least 3 cm (one to three samples per species), cut with a sliding microtome, permanently fixed, and analyzed under a microscope[56].

Species richness of heterotrophic organisms was assessed with a range of different methods between 2008 and 2012. We considered 11 groups of arthropods in our analyses, which we assigned to different functional groups according to their feeding ecology and trophic rank: parasitoids (parasitic wasps [Hymenoptera: Braconidae, Chrysididae, Eurytomidae, Ichneumonidae, Leucospidae, Mutillidae, Pompilidae, Trigonalyidae]), predators (spiders [Arachnida: Araneae], centipedes [Chilopoda], cavity-nesting solitary wasps [Hymenoptera: Pompilidae, Sphecidae, Vespidae], and strictly predatory as well as omnivorous ants [Hymenoptera: Formicidae]), primary consumers/herbivores (moth and butterfly caterpillars [Lepidoptera], weevils [Coleoptera: Curculioninae], longhorn beetles [Coleoptera: Cerambycidae], and bark beetles [Coleoptera: Scolytinae]) and decomposers (millipedes [Diplopoda] and isopods [Isopoda]). Epigeic spiders, centipedes, epigeic ants, weevils, isopods, and diplopods were sampled with pitfall traps[38] (four traps per plot from March to September 2009). Lepidopteran larvae, arboreal spiders, and ants were sampled from understory trees and shrubs by means of beating[57] (25 saplings per plot on three sampling dates in 2011 and 2012). Cavity-nesting predatory wasps and associated parasitic wasps were sampled with reed-filled trap nests (Supplementary Table 1). Longhorn beetles, bark beetles, and canopy ants were sampled with flight interception traps (4 traps per plot from May to August 2010). In addition, ants were sampled with standardized protein and carbohydrate baits[58] (36 baits per plot in May 2012). All arthropods were identified to species or morphospecies. Data on soil fungi were obtained from soil cores[20] (eight cores of the upper 10 cm of soil per plot, taken in September 2012). The soil was sieved and freeze-dried for molecular analysis. Fungal DNA was extracted with the MoBio soil DNA extraction kit and analyzed by pyrotag amplicon sequencing of the fungal internal transcribed spacer (ITS). Sequence datasets were quality filtered, normalized and clustered into species-level operational taxonomic units (OTUs). Non-target taxa OTUs as well as singletons, doubletons, and tripletons were removed from the dataset. The fungal reference sequences were assigned to ecological functional groups (saprophytes, pathogens, ectomycorrhizae, arbuscular mycorrhizae) on the basis of sequence similarity using the default parameters of the GAST algorithm[59] against the functional reference dataset[60].

**Abiotic attributes**. We measured a range of spatial and environmental variables in the study plots that might directly or indirectly influence ecosystem functions. Elevation (m above sea level), slope (°), degree of northness and eastness (cosine- and sine-transformed radian values of aspect), latitude, and longitude were assessed during plot establishment in 2008. Soil pH (measured potentiometrically in a $H_2O$ suspension), soil nitrogen (N) and carbon (C) contents (measured with Vario ELIII elemental analyzer, Elementar, Hanau, Germany), and the soil C:N ratio were determined from a bulk sample of nine soil cores (0–10 cm) per plot, taken in summer 2009. Mean annual temperature and mean January and July temperatures

per plot were obtained from continuous measurements with HOBO data loggers (one data logger in the center of each plot; 30 min time intervals from July 2011 to June 2012).

**Calculation of multifunctionality**. Each of the 22 measurements of ecosystem functions were scaled to range from 0 to 1 with the formula $f(x) = (x_i - x_{min})/(x_{max} - x_{min})$, where $x$ is the variable of interest with its minimum ($x_{min}$) and maximum ($x_{max}$) values observed across all study plots (see Supplementary Table 12 for details on observed values). Data on herbivory and erosion were multiplied by −1 prior to scaling to represent herbivory resistance and erosion control. All scaled measurements of a given function were then averaged per plot to obtain an ecosystem function variable that represents the mean of the various independent measurements, giving each function the same weight in the multi-functionality analyses. This yielded nine variables corresponding to the functions described above, on which all analyses were conducted.

From the various methods available to calculate multifunctionality, we chose two of the most commonly used: the "averaging approach" and the "multiple threshold approach"[5]. The averaging approach takes the mean value across all standardized functions as an index of multifunctionality for each study plot[5]. Threshold approaches measure how many functions simultaneously exceed a predefined percentage of the maximum observed value of each individual function[9,17,18]. As the selection of a given threshold is arbitrary, analyzing multiple thresholds of maximal functioning is recommended[5]. Thus, we used thresholds of 30%, 60%, and 90% to analyze how diversity affects multifunctionality at low, medium, and high levels, respectively, of the observed maximum functioning (see refs. [7,19] for similar approaches). We used the mean of the three largest values of each function as the observed maximum to reduce the impact of potential outliers[5]. The number of functions surpassing a given threshold was calculated with the R-package "multifunc".

**Trophic-level species richness and community composition**. We calculated a set of predictors that aggregated the information available for the diversity and community composition of woody plants and for the diversity of heterotrophic organisms at higher trophic levels (see Supplementary Table 12 for details on observed values).

From the woody plant inventory of each study plot, we calculated species richness and species composition of woody plants. Species composition was expressed as the first two principal components (PCs) of a principal components analysis (PCA) on relative abundance data, following the approach of ref. [39]. PCAs were conducted with the R package "vegan". Based on the relative abundance of the woody plant species, their morphological and chemical leaf traits, and their anatomical wood traits (see above), we quantified leaf morphological diversity, leaf chemical diversity, and wood anatomical diversity of the forest stands. We used Rao's quadratic entropy $Q$[61] for this, which describes the variance in pairwise trait dissimilarities among all individuals in a community. In addition, we calculated the community weighted mean values (CWMs) of the leaf and wood traits, i.e., the mean trait values across all individuals of a community. CWMs were subjected to a PCA for dimensionality reduction, yielding three PCs that accounted for the mean trait composition of the woody plant communities (Supplementary Table 6). Rao's $Q$ and CWMs were calculated with the R-package "FD". To control for confounding effects of the stand structure on diversity and composition effects, we included stand age and the density of woody plants in the study plots as further predictors.

To obtain metrics of heterotrophic species richness that are comparable across trophic levels and that allow combining data assessed with different sampling methods, we followed the approach of ref. [62] and calculated a diversity index that merges the species richness patterns of individual taxa. This was done by averaging the scaled (scaled to the maximum observed value across the study plots) species richness of the taxa in each functional group (i.e., parasitoids, predators, herbivores, macrofaunal decomposers, saprophytic fungi, parasitic fungi, mycorrhizae).

Finally, we included information on the environmental plot conditions as predictors, as they might have an influence on the ecosystem functions considered in our study[63]. The environmental variables were subjected to a PCA and we retained the first two PCs as a characterization of elevational differences in aboveground temperature (PC1) and belowground soil conditions (PC2; Supplementary Table 13).

**Effects of trophic-level diversity on multifunctionality**. We tested for multi-collinearity among the diversity and community composition variables and excluded the first PC of woody plant composition from the analyses (highly correlated with leaf chemical diversity, Pearson's $r = 0.78$, $P < 0.001$), and the second PC of trait composition (highly correlated with leaf morphological diversity, Pearson's $r = 0.72$, $P < 0.001$).

We used multiple linear regression on plot-level data with either the average multifunctionality index or the number of functions larger than the selected threshold as response variables. For the regression analyses, we used a model selection and averaging approach that calculated all possible subset models with up to four predictors and chose from this set those subset models with the lowest

values (ΔAICc ≤ 2) of the Akaike Information Criterion corrected for small sample size (AICc). As potential predictors, we considered diversity, composition, stand age, and environmental variables described above. We also considered potential interactions between stand age and diversity metrics in those cases where an initial data check indicated the possibility of significant or close to significant interactions (decomposer species richness × stand age, predator average species richness × stand age) and in the case of our design variable (tree species richness × stand age). For the selected set of models we used model averaging to address model uncertainty[64]. Model selection and multimodel averaging were conducted with the R-package "MuMIn". Adjusted $R^2$-values and partial $R^2$ values for individual predictors were averaged over all selected models. Model residuals were checked for normality and homogeneity of variances.

In addition to the analyses of multifunctionality based on the full set of functions and predictors across trophic levels, we conducted two further analyses to tease apart diversity effects within trophic levels vs. effects across trophic levels. As results for the above analyses on average and threshold multifunctionality showed the same general effects, we restricted the additional analyses to average multifunctionality for simplicity.

First, we calculated an overall index of total community richness as the average of the standardized diversity indices of (a) all trophic levels (average total community richness) and (b) all heterotrophic levels (average heterotrophic community richness) considered in our study. We re-ran the regression analyses as described above (AICc model selection and averaging based on a maximum of four predictors per model), but with one of the two community richness indices (and its interaction with stand age) instead of the individual diversity indices of the individual trophic levels as a predictor. Analogous to the regression analyses described above, environmental variables and plant-based predictors were considered as covariates. This analysis sheds light on the extent to which overall biodiversity (instead of the diversity of individual trophic levels) influences multifunctionality.

Second, we re-ran the analyses by trophic level (parasitoids, predators, herbivores, plants, decomposers, and fungi—the latter were considered together because functions such as microbial biomass and activity are the outcome of the combined effects of different soil fungi). For each trophic level, we excluded those functions from the multifunctionality index that were directly mediated by this trophic level (parasitoids: parasitism; predators: predation; herbivores: herbivory resistance; plants: primary productivity; decomposers: leaf and wood decomposition; fungi: microbial activity, nitrogen cycling, decomposition), as these functions might disproportionally influence the relationship between multifunctionality and diversity[65,66]. As predictors, we only considered the species richness of the respective trophic level, as well as environmental predictors and general plot characteristics (stand age, tree density) as covariables. These analyses allowed us to assess the extent to which diversity effects of a given trophic level propagate though the food web to affect ecosystem multifunctionality. We note that these comparisons across trophic levels are interdependent, which we cannot completely avoid with our relatively small sample size[67,68]. Results therefore need to be interpreted with care (see Results section for further details).

Finally, we used the same regression and model averaging approach with each of the nine individual ecosystem functions as a response variable and the set of environmental variables and the diversity metrics of all trophic levels as predictors. This allowed us to compare the multifunctionality results to the performance of individual ecosystem functions.

**Path analysis**. We used path analysis[69] to shed light on potential causal direct and indirect pathways that determine the combined effects of multiple metrics of biodiversity across trophic levels. The path model was informed by the outcome of the regression analysis of average multifunctionality. We included all pathways from plant-based predictors (exogenous variables, the three variables retained in the model had correlations with Pearson's $r \leq 0.2$) to heterotrophic predictors and multifunctionality (endogenous variables), as well as pathways between heterotrophic predictors and from these predictors to multifunctionality. We did not perform any model simplification and assessed model fit based on $\chi^2$-statistics. Models were fitted with the "lavaan" R-package.

All analyses were conducted in R 3.3.1 (www.r-project.org).

**Data availability**. All taxon data is available on the BEF-China project database. The pyrosequencing dataset of soil fungi is deposited in the EMBL SRA database under study number PRJEB8979.

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

## Acknowledgements

We thank the Administration Bureau of the Gutianshan National Nature Reserve and members of the BEF-China consortium for support, the many people involved in coordination and sampling (in particular M. Baruffol, M. Böhnke-Kammerlander, S. Both, N. Castro, C. Geißler, T. Fang, Y. Huang, W. Kröber, Z. Pei, X. Yang, and P. Zumstein). We thank two anonymous reviewers for constructive comments that helped to improve the manuscript. We gratefully acknowledge funding by the German Research Foundation (DFG FOR 891/1–3), the Sino-German Centre for Research Promotion (GZ 524, 592, 698, 699, 785, and 1020), and the National Natural Science Foundation of China (NSFC 30710103907 and 30930005). We acknowledge the financial support of the Open Access Publication Fund of the Martin-Luther-University Halle-Wittenberg. A.S. acknowledges support by the German Centre for Integrative Biodiversity Research (iDiv) Halle-Jena-Leipzig (DFG FZT 118). In memory of Matteo Brezzi, who tragically passed away in March 2018.

## Author contributions

A.S. conceived the idea for the manuscript. H.B., K.M., B.S., C.W., T.A., F.B., J.G., W.H., A.M.K., P.K., P.A.N., M.S.L., A.S., T.S., G.v.O., and T.W. designed research. M.B., D.E., J.G., J.S.H., X.L., K.A.P., W.P., A.S., M.S., Z.T., S.T., T.W., and C.D.Z. collected and/or contributed data. A.S. conducted the statistical analyses and wrote the manuscript, with input from all coauthors.

## Additional information

**Competing interests:** The authors declare no competing interests.

