## [Peer Review File · Nature Communications]

Reviewers' comments:

Reviewer #1 (Remarks to the Author):

This study investigates how biodiversity influences multifunctionality in highly diverse subtropical forests in China. The results are based on 27 plots from five successional stages (<20, <40, <60, <80, and >80 years). The work builds upon recent work published in key journals (e.g. Nature, Science, PNAS) investigating multifunctionality in forests and studies that test how multitrophic diversity influence ecosystem multifunctionality. In this paper, a large amount of data are being presented and analysed.

While I do like the approach, I have three main comments. 1) It is specifically tested whether biodiversity (and tree richness) is influencing multifunctionality. However, the selected plots represent a successional gradient and I wonder whether it would not be more relevant to test the effects of succession on multifunctionality as well. It could be tested how successional stage influences the observed relationships and the interactive effects of succession and diversity could be discussed in more detail. 2) The presented results present "aggregated" results and no "real" data. It is important, from my point of view to also show the actual data. Would it be possible to show scatterplots for all the relationships presented in Figure 2. In such a figure, values for the different successional stages should be labelled differently. Similarly, Figure 1 lacks any data points and instead of standardized data it would be useful to also present the actual data (e.g. the observed woody plant species richness, etc.). Again, it would be useful to differently label the successional stages. 3) In recent years a number of papers have been published linked to the forest under study. It would be useful to present an overview of all these papers and which aspects are covered in each of these paper (this can be a table in the Supplement – to make clear what is new about this work).

Specific comments:

Line 64: is this the first study testing whether multitrophic diversity influences ecosystem multifunctionality for tropical forests? If so, it is important to stress this here clearly.

L134: please remove "also"

Line 136: it is important to show the actual data for individual variables/ functions.

L195: previous studies highlighted the importance of tree species richness for forest multifunctionality. In this study, a negative relationship was observed. It needs to be explained why this is. Probably this is linked to the fact that the richness gradient is dependent on successional stage.

L209: A range of important relationships (positive and negative correlations) between variables are presented (e.g. diversity of heterotrophic groups of organisms contributed significantly to explaining changes in multifunctionality). With so many significant correlations it is always difficult to find the main driver. Is it possible to perform an analysis that is able to detect the main drivers and the underlying processes (e.g. structural equation modelling is an option, but the number of plots is probably too low for this).

Table 1 and 2: please also present R² levels for the individual predictors in the table.

Supplementary table 1 summarize nicely all the variables being analysed and the methods used to analyse them.

Reviewer #2 (Remarks to the Author):

The study by Schuldt et al is a very nice and comprehensive evaluation of the effects of multitrophic biodiversity on multifunctionality. This work reports on one of the richer datasets on ecosystem functioning that I have yet seen: generally authors addressing multifunctionality have to split hairs as to what constitutes a function, whereas the authors here have actually *collapsed* 22 functions into

9. This is a smart decision as recent work has shown that correlations among functions may influence some of these multifunctionality indices.

However, as reading, I am pressed to find the novelty here. There has been a deluge of multifunctionality papers recently (some applications, some critical reevaluations) and I am struggling to see what sets this paper apart. Especially since the central thesis they propose is that diversity of higher trophic levels (e.g., above primary producers) has rarely been considered within the context of multifunctionality, yet one of the primary messages of the 2015 meta-analysis by Lefcheck et al (cited within) is that herbivore diversity enhances multifunctionality to a greater degree than do primary producers or decomposers. Granted, this is a far more comprehensive demonstration within a relatively understudied realm (tropical forests), but I'm not sure that is the ideal pitch for this work. The larger issue I have is that, for a multitrophic paper, the analysis is still highly compartmentalized. For example, the authors note that belowground diversity influences belowground functions most strongly. This is not surprising as there is a much more convoluted pathway to explain why, for example, predator diversity might influence decomposition. So, I wonder how much of this multifunctionality effect is being driven by positive relationships *within* trophic levels vs *between* trophic levels? (the latter being the more interesting contribution and most directly relevant to the central thesis of the manuscript.)

Based on Figure 2, my suspicion is that the overall trends are biased by the inclusion of these strong within-trophic level correlations that are leading the authors to infer a stronger multitrophic signal than actually exists. Recent work Gamfeldt & Roger 2017 and Meyer et al 2017 (both in Nature Ecology & Evolution) suggest that these indices are sensitive to both the identity and number of functions, so that they can in essence be 'hijacked' by a few functions that are positively correlated with diversity.

I offer two potential solutions: (1) re-run the analysis by trophic level, but omit the functions promoted by that trophic level from the index of multifunctionality; or alternately (2) collapse diversity into total community diversity. The former will allow you to assess linkages that permeate across the food web, the latter will address the question of whether biodiversity writ large matters for multifunctionality writ large. Both of these outcomes will speak more to multitrophic biodiversity effects than the current implementation.

I will admit that there are some interesting effects between higher trophic level diversity and functions related to primary producers, however, some of these are counter intuitive (high herbivore diversity = *higher* functioning of primary producers?). A more thorough investigation and discussion of these relationships might further emphasize the role of diversity across trophic levels. The expectations are tricky: see review by Duffy et al. 2007 in Ecology Letters and papers cited within.

Again, I think there is tremendous promise in the data. I personally would love to work with such a well-resolved system in terms of functioning! I think the question is relevant, and the execution is competent, but I think there are some underlying issues with the metrics and potential mismatching of ideas that is giving me pause. I trust (and hope!) the authors can extract some additional and interesting trends from the data!

A few minor comments:

-I am seriously worried the authors are overfitting their models. They have 27 data points and 18 predictors?! I see the application of model selection has whittled this down to 8 but having recently attempted to apply linear models with a similar sample size, I worry that Table 1 is potentially misleading. The general rule of thumb is 10-15 observations per predictor so at most the authors could squeeze 3-4 predictors into a model. Suggestion #1 above might be useful in this regard HOWEVER be wary about multiple testing and be sure to adjust alpha accordingly

-the use of the term 'multidiversity' is totally misleading. As far as I can tell, it's proportional richness. It doesn't integrate across multiple trophic levels, as might be erroneously inferred from the title. I recommend calling this simply 'diversity' or 'proportional richness,' if you wish to be more precise

Reply to Reviewers' comments:

Reviewers' comments:

Reviewer #1 (Remarks to the Author):

This study investigates how biodiversity influences multifunctionality in highly diverse subtropical forests in China. The results are based on 27 plots from five successional stages (<20, <40, <60, <80, and >80 years). The work builds upon recent work published in key journals (e.g. Nature, Science, PNAS) investigating multifunctionality in forests and studies that test how multitrophic diversity influence ecosystem multifunctionality. In this paper, a large amount of data are being presented and analysed.

While I do like the approach, I have three main comments.

1) It is specifically tested whether biodiversity (and tree richness) is influencing multifunctionality. However, the selected plots represent a successional gradient and I wonder whether it would not be more relevant to test the effects of succession on multifunctionality as well. It could be tested how successional stage influences the observed relationships and the interactive effects of succession and diversity could be discussed in more detail.

- Thank you for your helpful comments. We agree that the age of the study plots is an important co-variable, and we had already included it in our initial analyses. Maybe this was not really evident because we used the estimated age of the study plots ('stand age' in the manuscript) as a continuous predictor instead of categorical successional stages (which were only used at the time of plot establishment and in early papers as very coarse estimates of successional age until a better estimate based on stem cores became available), because stand age provides a more accurate estimate of the real successional age of the study plots. However, we agree that potential interactions between diversity metrics (in particular with our design variable 'woody plant species richness') and stand age should be considered as well. Therefore, we re-analyzed our data and included potentially relevant diversity x stand age interactions in the model selection procedure (while also considering the concerns of Reviewer 2 with respect to potential overfitting of models). We changed the text in the Methods accordingly (L526-531):

"As potential predictors, we considered diversity, composition, and environmental variables described above. We also considered potential interactions between stand age and diversity metrics in those cases where an initial data check indicated the possibility of significant or close to significant interactions (decomposer diversity x stand age, predator diversity x stand age) and in the case of our design variable (tree species richness x stand age)."

None of the models selected in the automatic AICc procedure contained stand age or a significant stand age x diversity interaction (except for the new additional analysis of overall multidiversity suggested by Reviewer 2). We have added this information to the relevant section in the Discussion (L319-324):

“However, the fact that the biotic attributes of the plant communities were stronger predictors of multifunctionality than abiotic variables, and that stand age or diversity interactions with stand age had only limited predictive power (as they were only retained in the final model of overall multidiversity but in none of the other models), suggests that changes in biotic attributes are key to mechanistically explaining multifunctionality.”

2) The presented results present aggregated results and no real data. It is important, from my point of view to also show the actual data. Would it be possible to show scatterplots for all the relationships presented in Figure 2. In such a figure, values for the different successional stages should be labelled differently. Similarly, Figure 1 lacks any data points and instead of standardized data it would be useful to also present the actual data (e.g. the observed woody plant species richness, etc.). Again, it would be useful to differently label the successional stages.

- We agree that scatterplots with data points can be useful, and we now provide residual plots for all relationships between diversity metrics and multifunctionality (Supplementary Fig. 4 for the data shown in Fig 4 (formerly Fig. 2), Supplementary Fig. 3 for threshold multifunctionality, new Figs 1 & 3 for average multifunctionality). The data points have been colored according to the stand age of the study plots (showing the lack of interactive effects between diversity and stand age for these relationships). For completeness, we provide the plots containing only the predicted relationships in Supplementary Figs 1 & 2 (without data points to increase the clarity of the presentation of several lines in each panel). Showing non-standardized data would only make sense for woody plant species richness, because all other diversity metrics either had to be standardized to calculate diversity indices (trophic level diversity—formerly termed ‘multidiversity’) or because they are dimensionless values derived from PCA reduction (compositional variables).

3) In recent years a number of papers have been published linked to the forest under study. It would be useful to present an overview of all these papers and which aspects are covered in each of these paper (this can be a table in the Supplement; to make clear what is new about this work).

- We have added this information as Supplementary Table 10, where we also discuss the innovation of our current manuscript in comparison to these previous studies. This Table should make clear that there is no real overlap between the current manuscript (diversity effects across multiple trophic levels on multifunctionality) and previous studies (effects of woody plant diversity on individual taxa or functions). We refer to this Table in the Methods (L390-393):

“Our study provides a comprehensive analysis of these functions in a multifunctional and multitrophic context, which differs substantially in scope from previous analyses in our study system that looked at individual functions and their relationship with woody plant communities (Supplementary Table 10).”

Specific comments:

Line 64: is this the first study testing whether multitrophic diversity influences ecosystem multifunctionality for tropical forests? If so, it is important to stress this here clearly.

- There have been few studies on multifunctionality in (sub)tropical forests so far and, as far as we know, ours is the first study to analyze diversity effects across multiple trophic levels (from soil microorganisms to aboveground arthropods) on a large set of ecosystem functions. We have added “*for the first time*” in L111.

L134: please remove “also”

- Done.

Line 136: it is important to show the actual data for individual variables/ functions.

- Done, see comment above.

L195: previous studies highlighted the importance of tree species richness for forest multifunctionality. In this study, a negative relationship was observed. It needs to be explained why this is. Probably this is linked to the fact that the richness gradient is dependent on successional stage.

- We now elaborate on this aspect in our Discussion section. . Please note that the richness gradient was not significantly affected by successional stage, because levels of woody plant species richness varied within successional stages. (Pearson’s $r = 0.24$, $P = 0.22$ for the correlation between woody plant species richness and stand age). We now also show that there were no significant diversity x stand age interactions in our analyses (see above). Moreover, based on the suggestions of Reviewer 2 (fewer predictors per model), our revised analyses do no longer show a significant negative effect of woody plant species richness (which in our previous analyses was only evident in the analysis of multifunctionality at a 30% threshold, but not in any of the other analyses, so that the overall effect of woody plant species richness was weak). In the Discussion, we have added (L328-337):

“The weak effects of woody plant species richness observed in our study contrast with findings of previous studies on multifunctionality in forests^{6,8-10}, but might be explained by the much higher woody plant species richness of our forest stands. Biodiversity effects are often particularly pronounced at lower levels of species richness, and our findings might reflect the often expected levelling-off of diversity effects at higher levels of species richness²⁵. Nevertheless, some of the functions we examined did show significant relationships with woody plant species richness (leaf decomposition, herbivory resistance). This indicates that while plant species richness can influence individual ecosystem functions even at high richness levels, these effects may not be strong enough to influence overall multifunctionality in our study system.”

L209: A range of important relationships (positive and negative correlations) between variables are presented (e.g. diversity of heterotrophic groups of organisms contributed significantly to explaining changes in multifunctionality). With so many significant correlations it is always difficult to find the main driver. Is it possible to perform an analysis

that is able to detect the main drivers and the underlying processes (e.g. structural equation modelling is an option, but the number of plots is probably too low for this).

- We have added a path analysis to show how the effects of plant and heterotrophic diversity indicated by the regression analyses are connected via direct or indirect interactions. We describe this procedure in the Methods (L571-579):

“We used path analysis⁷⁰ to shed light on potential causal direct and indirect pathways that determine the combined effects of multiple metrics of biodiversity across trophic levels. The path model was informed by the outcome of the regression analysis of average multifunctionality. We included all pathways from plant-based predictors (exogenous variables, the three variables retained in the model had correlations with Pearson’s $r \leq 0.2$) to heterotrophic predictors and multifunctionality (endogenous variables), as well as pathways between heterotrophic predictors and from these predictors to multifunctionality. We did not perform any model simplification and assessed model fit based on Chi-square statistics. Models were fitted with the lavaan R-package.”

In the Results (L145-151) and Discussion (L244-246) we elaborate on these interconnections. We have added a new figure (Fig. 2) that summarizes the results of the path analysis (statistical output is provided in Supplementary Table 5).

Table 1 and 2: please also present R² levels for the individual predictors in the table.

- We have added partial R² values for each predictor as requested (see Supplementary Tables 2, 3, 7, 8, 9).

Supplementary table 1 summarize nicely all the variables being analysed and the methods used to analyse them.

- Thank you.

Reviewer #2 (Remarks to the Author):

The study by Schuldt et al is a very nice and comprehensive evaluation of the effects of multitrophic biodiversity on multifunctionality. This work reports on one of the richer datasets on ecosystem functioning that I have yet seen: generally authors addressing multifunctionality have to split hairs as to what constitutes a function, whereas the authors here have actually ***collapsed*** 22 functions into 9. This is a smart decision as recent work has shown that correlations among functions may influence some of these multifunctionality indices.

- Thank you.

However, as reading, I am pressed to find the novelty here. There has been a deluge of multifunctionality papers recently (some applications, some critical reevaluations) and I am

struggling to see what sets this paper apart. Especially since the central thesis they propose is that diversity of higher trophic levels (e.g., above primary producers) has rarely been considered within the context of multifunctionality, yet one of the primary messages of the 2015 meta-analysis by Lefcheck et al (cited within) is that herbivore diversity enhances multifunctionality to a greater degree than do primary producers or decomposers. Granted, this is a far more comprehensive demonstration within a relatively understudied realm (tropical forests), but I'm not sure that is the ideal pitch for this work.

- We agree that the analysis of multifunctionality–diversity relationships is currently featured very prominently in many interdisciplinary journals, such as *Nature Ecology & Evolution*. This highlights the importance of this topic and the wealth of papers shows that there are many issues that have not been resolved so far. We agree that the Lefcheck et al. meta-analysis of experimental manipulations provides an important general framework and overview of higher trophic level effects on multifunctionality, and in our revised manuscript we highlight this contribution by adding citations to the meta-analysis early in the Introduction (L80, 84). Nevertheless, our study of a highly diverse, near-natural ecosystem contributes to addressing important knowledge gaps that, in combination, have not been adequately addressed so far. In particular, we address i) the relative importance of different facets of biodiversity and community composition for ecosystem multifunctionality, ii) the influence of higher trophic level multidiversity, and iii) the transferability of current knowledge to highly diverse subtropical and tropical ecosystems. Previous studies have largely focused on the effects of plant species richness on ecosystem multifunctionality in temperate and boreal ecosystems, but neither on these effects in subtropical forest ecosystems nor on effects of “multidiversity” on ecosystem functioning. As such, we think that our study provides important information to move the field forward.

However, we are very grateful for the suggestions of the Reviewer regarding the additional analyses, which helped us to further strengthen our approach by more clearly identifying the role of individual higher trophic levels (i.e. effects within vs. between trophic levels on multifunctionality) and of multidiversity as a whole, which has not been done in this way in multifunctionality analyses so far. We have incorporated this information also in the Introduction and Conclusions of our manuscript to distinguish our work even better from previous studies, e.g. L125-129:

“In addition, we tested the extent to which effects of higher trophic-level diversity on multifunctionality propagate through the food web by analyzing the contribution of the diversity of individual trophic levels to multifunctionality after excluding functions directly modified by a given trophic level, and by testing for effects of overall diversity across trophic levels.”

And e.g. L211-213:

“Importantly, the effects of higher trophic-level diversity propagated through the food web, indicating that higher trophic levels have important indirect effects on ecosystem functions driven by adjacent trophic levels.”

The larger issue I have is that, for a multitrophic paper, the analysis is still highly compartmentalized. For example, the authors note that belowground diversity influences belowground functions most strongly. This is not surprising as there is a much more convoluted pathway to explain why, for example, predator diversity might influence decomposition. So, I wonder how much of this multifunctionality effect is being driven by

positive relationships ***within*** trophic levels vs ***between*** trophic levels? (the latter being the more interesting contribution and most directly relevant to the central thesis of the manuscript.)

Based on Figure 2, my suspicion is that the overall trends are biased by the inclusion of these strong within-trophic level correlations that are leading the authors to infer a stronger multitrophic signal than actually exists. Recent work Gamfeldt & Roger 2017 and Meyer et al 2017 (both in Nature Ecology & Evolution) suggest that these indices are sensitive to both the identity and number of functions, so that they can in essence be hijacked; by a few functions that are positively correlated with diversity. I offer two potential solutions: (1) re-run the analysis by trophic level, but omit the functions promoted by that trophic level from the index of multifunctionality; or alternately (2) collapse diversity into total community diversity. The former will allow you to assess linkages that permeate across the food web, the latter will address the question of whether biodiversity writ large matters for multifunctionality writ large. Both of these outcomes will speak more to multitrophic biodiversity effects than the current implementation.

- Thank you for these very interesting suggestions. We have followed your recommendations and have added both proposed analyses to our manuscript. By comparing the results to our overall analysis of multifunctionality (i.e. the initial analyses, which we kept because they are important to show effects on overall multifunctionality), we can show that diversity effects of individual trophic levels are not only important when functions directly mediated by a given trophic level are included. In fact, most trophic level-effects propagate through the food web to influence functioning at other trophic levels. Decomposers are an exception, as our new analyses show that their effects on multifunctionality are particularly strongly driven by their impact on decomposition functions, whereas effects across the food web are only marginally significant. We can also show that overall biodiversity (the average of the standardized diversity index across trophic levels) positively influences multifunctionality (especially in older forest stands), which further supports our finding that heterotrophic diversity needs to be more thoroughly considered in multifunctionality studies.

We have added a description of the additional analyses to the Methods (L536-564):

"In addition to the analyses of multifunctionality based on the full set of functions and predictors across trophic levels, we conducted two further analyses to tease apart diversity effects within trophic levels vs. effects across trophic levels. Because results for the above analyses on average and threshold multifunctionality showed the same general effects, we restricted the additional analyses to average multifunctionality for simplicity. First, we re-ran the analyses by trophic level (parasitoids, predators, herbivores, plants, decomposers, and fungi—the latter were considered together because functions such as microbial biomass and activity are the outcome of the combined effects of different soil fungi). For each trophic level, we excluded those functions from the multifunctionality index that were directly mediated by this trophic level (parasitoids: parasitism; predators: predation; herbivores: herbivory resistance; plants: primary productivity; decomposers: leaf and wood decomposition; fungi: microbial activity, nitrogen cycling, decomposition), because these functions might disproportionately influence the relationship between multifunctionality and diversity^{66,67}. As predictors, we only considered the diversity of the respective trophic level, as well as environmental predictors and plant-based diversity and composition (because they might mediate higher trophic-level effects) as covariables. These analyses allowed us to assess the extent to which diversity effects of a given trophic level propagate through the food web to affect ecosystem multifunctionality. We note that these comparisons across trophic levels are interdependent, which we cannot completely avoid

with our relatively small sample size^{68,69}. Results therefore need to be interpreted with care (see Results section for further details).

Second, we calculated an overall index of multidiversity as the average of the standardized diversity indices of all trophic levels considered in our study. We re-ran the regression analyses as described above (AICc model selection and averaging based on a maximum of four predictors per model), but with the multidiversity index (and its interaction with stand age) instead of the individual diversity indices of heterotrophic organisms as a predictor. Environmental variables and plant-based predictors were considered as covariates. This analysis sheds light on the extent to which overall biodiversity (instead of the diversity of individual trophic levels) influences multifunctionality.”

The outcome of these analyses is reported in a new figure (Fig. 3) and a new section in the Results (L177-194):

“Linkages across trophic levels and multidiversity effects on multifunctionality

We also found significant diversity effects on multifunctionality when excluding functions directly related to a given trophic level. Four of the six trophic levels showed significant diversity effects that were not dependent on functions directly mediated by the respective trophic level, indicating that these effects propagate through the food web (Fig. 3, Supplementary Table 7). This was also the case for the diversity of parasitoids and herbivores, whose effects on multifunctionality were masked by stronger diversity predictors in the overall analyses when the diversity of all functional groups was considered simultaneously. The effects of decomposer diversity on multifunctionality after excluding leaf and wood decomposition depended on stand age and were only marginally significant (Fig. 3, Supplementary Table 7). Because these comparisons are interdependent, significance levels need to be interpreted with care. Nevertheless, significant diversity effects in four of the models (with $\alpha = 0.05$ we would expect only one out of the six comparisons to show a significant effect by chance) and the results of the overall model above support the biological relevance of our results.

Multidiversity, as the average of the standardized diversity indices across trophic levels, positively affected ecosystem multifunctionality in older forest stands, but not in young forest stands (Fig. 3, Supplementary Table 8). “

The implications of these findings are discussed throughout the Discussion section (e.g. L211-213):

“Importantly, the effects of higher trophic-level diversity propagated through the food web, indicating that higher trophic levels have important indirect effects on ecosystem functions driven by adjacent trophic levels.”

And L239-246:

“Macrofaunal decomposer diversity was positively related to multifunctionality, in accordance with the assumption that more diverse decomposer communities contribute more effectively to decomposition processes³¹. This biodiversity–multifunctionality relationship was strongly influenced by the inclusion of leaf and wood decomposition, indicating that effects of decomposer diversity are less likely to cascade up to functions mediated by higher trophic levels (e.g. parasitism). However, path analyses showed that indirect effects of woody plant composition on multifunctionality via decomposer diversity weakened direct plant effects on multifunctionality.”

I will admit that there are some interesting effects between higher trophic level diversity and functions related to primary producers, however, some of these are counter intuitive (high herbivore diversity = ***higher*** functioning of primary producers?). A more thorough investigation and discussion of these relationships might further emphasize the role of diversity across trophic levels. The expectations are tricky: see review by Duffy et al. 2007 in Ecology Letters and papers cited within.

- We agree that the analysis of individual functions reveals interesting insights into specific relationships between trophic level diversity and individual functions that contribute to the effects of heterotrophic diversity on multifunctionality. Based on the suggestions below to reduce the number of predictors in our models (from max. 8 to max. 4), some of these effects are not evident any more. This also applies to the above-mentioned positive effects of herbivore diversity on primary productivity. However, in our case such a relationship might be explained by the fact that herbivory (and therefore negative effects on productivity) was strongly determined by generalist herbivores. A higher diversity of herbivores might reduce the dominance of these generalist herbivores and subsequently the potentially negative effects of these herbivores on productivity. The other relationships between higher trophic level diversity and ecosystem functions at the plant and other trophic levels are more intuitive and are discussed in the text (e.g. for decomposers and saprophytic fungi: L235-259). As another example to strengthen the discussion of heterotrophic diversity effects, we have added the case of predator diversity on primary productivity, as suggested by the Reviewer (L274-281):

“Including a wider set of ecosystem functions is important for sustainable forest management particularly in the face of climate change-induced management challenges, such as increased frequency and intensity of insect herbivore outbreaks¹⁶. In this context, the strong, positive effect of predator diversity on primary productivity observed in our study—indicative of increasing top-down control of herbivores—is highly relevant. Strengthening resistance against herbivores and biocontrol by predators and parasitoids may contribute to developing suitable strategies that ensure a stable provisioning of ecosystem services under uncertain future environmental conditions.”

Again, I think there is tremendous promise in the data. I personally would love to work with such a well-resolved system in terms of functioning! I think the question is relevant, and the execution is competent, but I think there are some underlying issues with the metrics and potential mismatching of ideas that is giving me pause. I trust (and hope!) the authors can extract some additional and interesting trends from the data!

- Thank you for these positive words. As explained in our reply to your comments above, we hope that the additional analyses and the changes made to the text help to resolve these issues.

A few minor comments:

-I am seriously worried the authors are overfitting their models. They have 27 data points and 18 predictors?! I see the application of model selection has whittled this down to 8 but having recently attempted to apply linear models with a similar sample size, I worry that Table 1 is potentially misleading. The general rule of thumb is 10-15 observations per predictor so at most the authors could squeeze 3-4 predictors into a model. Suggestion #1 above might be

useful in this regard HOWEVER be wary about multiple testing and be sure to adjust alpha accordingly

- For the revised manuscript, we have followed this suggestion and have included a maximum of four predictors per model, both in the overall approach and in the additional approach suggested by the Reviewer. We have also clarified our statistical approach in the Methods to avoid potential misunderstanding. That way, it should be clear that we are not fitting 18 predictors in one model. However, because our model averaging approach makes use of multiple models to address model uncertainty, the overview presented in the tables can contain more than four predictors (because the models selected for the averaging approach will differ to some extent in the identity of the predictors). The information provided in the tables (e.g. on the number of models used for model averaging) should clarify this aspect.

We now write in the Methods section (L523-532):

“For the regression analyses, we used a model selection and averaging approach that calculated all possible subset models with up to four predictors and chose from this set those subset models with the lowest values ($\Delta AICc \leq 2$) of the Akaike Information Criterion corrected for small sample size (AICc). As potential predictors, we considered diversity, composition, and environmental variables described above. We also considered potential interactions between stand age and diversity metrics in those cases where an initial data check indicated the possibility of significant or close to significant interactions (decomposer diversity x stand age, predator diversity x stand age) and in the case of our design variable (tree species richness x stand age). For the selected set of models we used model averaging to address model uncertainty⁶⁵.”

-the use of the term ‘multidiversity’ is totally misleading. As far as I can tell, it’s proportional richness. It doesn’t integrate across multiple trophic levels, as might be erroneously inferred from the title. I recommend calling this simply ‘diversity’ or ‘proportional richness,’ if you wish to be more precise

- We agree that the term multidiversity might not be very intuitive when it refers to the average diversity of multiple groups of organisms within a trophic level (as in our study). We have now restricted this term to overall multidiversity (i.e. the standardized average of diversity across trophic levels) in the additional analyses suggested by the Reviewer. In all other cases (including in the title), we now use “higher trophic level diversity” instead.

Reviewers' comments:

Reviewer #1 (Remarks to the Author):

The authors addressed several of my comments. I still do have a number of comments.

Summary: the summary states that multifunctionality is more affected by the diversity of heterotrophs and by plant functional trait diversity than by tree species richness. However, it never states that biodiversity is positively related to multifunctionality (which the title says). Please clarify this. The results in Figure 3 suggest that there is no relationship between the overall diversity (multi-diversity) and multifunctionality. If that is true, the title needs to be changed.

Figure 3 shows the relationship between diversity and functionality (the figure legends says it is functionality but I assume it must be multifunctionality)? The bottom right panel show the relationship between "overall diversity" and functionality. Is this multi-diversity (as mentioned in the methods)? The relationship for young forests in that figure is significant according to the figure (solid line) and not significant according to the text (line 194). Please modify. Please also present the overall relationship between diversity and multifunctionality for all plots (young, middle and old). Is that relationship significant?

The methods states that the species richness in each functional group are taken to calculate multi-diversity (L505). However, plant species richness is not included. Plant species richness should also be included because plant species richness contributes to the "overall diversity" of a forest plot. Also, it is better to call this richness (and not diversity) if only richness data are used. Would it be possible to calculate the effects of multi-diversity on ecosystem multifunctionality with and without plant/woody richness data included.

Apparently there is no effect of succession and successional age on multifunctionality (see line 319-323 & Authors response to my queries about this). I am a bit surprised and find it hard to imagine that tree species of the young sites have similar characteristics as the tree species of the old sites? There are many examples in the ecological literature which show that early successional species are different from late successional species.

Following the logic of having no "successional effect": Is it correct that in these forest no typical "early successional" and "late successional" species can be found (with early successional species, I mean species that colonize a disturbed habitat, that could even be a patch within a late successional forest). I would not be surprised if biological variables (which are a consequence of successional age) overrule and mask the importance of successional age.

It should be stated somewhere that this study describe a range of forest plots. Thus, the results presented are descriptive and in the end correlations among a wide range of variable are presented. Hence, it is important to state somewhere that such relationship are not based on causality and other underlying factors could be responsible for observed effects and relationships (e.g. is decomposer diversity really running the system, in terms of multifunctionality).

On my request (see authors response), the authors now present residual scatterplots with data points. Please present the real data (no residual or transformed data). This will make it possible for readers and other researchers to compare values of this study with other studies (e.g. for instance, tree species richness of these forests or productivity or the number of decomposer species). Also, with residual data it is not possible (as reviewer) to verify whether the numbers reflect what one would typically expect for the studied forests.

Line 178-179: I found this line unclear. What is meant exactly? Please add another sentence to clarify the goals.

Line 439: ITS58 does not exist. Please modify.

Line 462: please rephrase sentence (is not completely clear): I assume x-min means the x-min of all studied plots?

Reviewer #2 (Remarks to the Author):

The authors have done a commendable job in addressing my initial concerns. While I did not suggest it, I appreciate the application of SEM to look at the cascading effects, as it makes a lot of sense and I'm glad to see its inclusion.

I was overall happy with the revised manuscript, but I was still a little confused by the presentation. In Figure 1, only 4 aspects of diversity are considered (woody plant, chemical, decomposer, fungi), while in Figure 3, parasitoid, herbivore, and predator diversity are additionally considered (while woody plant diversity is not). The mismatch is not explained in the text nor is it accounted for in the supplementary figures. I would instead present all aspects of diversity at once.

A more effective way to structure these results would be to have a single two-column figure, one with different aspects of diversity against multifunctionality (e.g., woody plant, etc.) and the second with the same diversity component against multifunctionality MINUS the functions related to that component. So, a combined figure 1 and 3. If this is unwieldy, the non-diversity components could be a different figure (e.g., PCA1 & 2, chemical diversity, etc.) since they are interpreted separately.

Finally, a key finding from this paper is that diversity of various trophic levels is often *negatively* correlated with multifunctionality (Figures 1, 3). Possible reasons for this are given for fungi (lines 246-259), but I find it strange that, for example, herbivore, predator, and parasitoid diversity is always neutrally or positively correlated with individual functions (Figure 4). Yet, these predictors are generally neutrally or negatively correlated with average multifunctionality (Figure 3). Some explanation as to why this is would be warranted in the discussion.

Overall, nice effort, I think with cleaning up these minor issues, the paper should be very well received.

Minor comments:

Abstract: should include a mention of the (new tests of) cascading effects

Lines 79-83: this is a bit of a strawman argument. While its true that some studies consider diversity to be a 'function,' many analyses are explicitly interested in it as a driver

Line 192: omit 'multidiversity' and call it 'total community richness'

Line 276-278: cite Supp Table 8 here, since this is not in the main figures

Your manuscript entitled "Biodiversity across trophic levels maximizes multifunctionality in highly diverse forests" has now been seen by 2 referees. You will see from their comments below that while they find your work of interest, some important points are raised. We are interested in the possibility of publishing your study in Nature Communications, but would like to consider your response to these concerns in the form of a revised manuscript before we make a final decision on publication.

We therefore invite you to revise and resubmit your manuscript, taking into account the points raised. Please highlight all changes in the manuscript text file.

- We would like to thank you and the reviewers again for the constructive review and for the time and effort invested in reviewing our manuscript! We are grateful for the additional comments and suggestions and have addressed all of them in our revised manuscript. For details, please see our detailed response to the reviewers' comments below.

At the same time, we ask that you ensure your manuscript complies with our editorial policies. Please ensure that the following requirements are met, and any relevant checklist is completed or updated and uploaded as a Related Manuscript file type with the revised article.

- We have thoroughly checked the editorial policies and provide all requested additional information together with our revised manuscript.

Reviewers' comments:

Reviewer #1 (Remarks to the Author):

The authors addressed several of my comments. I still do have a number of comments.

- Thank you for your time and valuable suggestions for improvement of the manuscript!

Summary: the summary states that multifunctionality is more affected by the diversity of heterotrophs and by plant functional trait diversity than by tree species richness. However, it never states that biodiversity is positively related to multifunctionality (which the title says). Please clarify this. The results in Figure 3 suggest that there is no relationship between the overall diversity (multi-diversity) and multifunctionality. If that is true, the title needs to be changed.

-We agree that our results do not simply show positive effects of biodiversity on multifunctionality, as relationships with the latter were negative for several functional groups (e.g. saprophytic fungi). We therefore changed the title and substituted "maximizes" by "drives" so that any misunderstanding is hopefully avoided. Regarding the comment on Figure 3, please see our reply below to the next comment.

Figure 3 shows the relationship between diversity and functionality (the figure legends says it is functionality but I assume it must be multifunctionality)? The bottom right panel show the relationship between overall diversity; and functionality. Is this multi-diversity (as mentioned in the methods)? The relationship for young forests in that figure is

significant according to the figure (solid line) and not significant according to the text (line 194). Please modify. Please also present the overall relationship between diversity and multifunctionality for all plots (young, middle and old). Is that relationship significant?

-Thank you for making us aware of these issues. We have updated the figure so that it is clear that multifunctionality is shown on the y-axis. The multidiversity~multifunctionality plot is now in a new figure (Fig. 4) and we have added information to the figure legend explaining the meaning of multidiversity (which you correctly assume was called “Overall diversity” in the previous manuscript version – and which we have now replaced by “Average total community richness” as the x-axis label, following the suggestion of Reviewer 2). We have taken care that the non-significant relationship in young forest plots is now indicated by a broken regression line. The overall relationship between total community richness and multifunctionality is now shown in a separate figure (Supplementary Figure 7b). In the text (Results, L204-208) we now explicitly mention that the effects of total community richness depended on stand age and that the main effects of total community richness and stand age were not significant.

The methods states that the species richness in each functional group are taken to calculate multi-diversity (L505). However, plant species richness is not included. Plant species richness should also be included because plant species richness contributes to the overall diversity of a forest plot. Also, it is better to call this richness (and not diversity) if only richness data are used. Would it be possible to calculate the effects of multi-diversity on ecosystem multifunctionality with and without plant/woody richness data included.

-Thank you for the suggestion on calculating multidiversity with and without woody plant species richness. Multidiversity in our previous manuscript version actually included woody plant species richness. We have revised the relevant section in the Methods to clarify this and to describe our new approach to analyze multidiversity both with and without plants (L570-573):

“First, we calculated an overall index of total community richness as the average of the standardized diversity indices of a) all trophic levels (average total community richness) and b) all heterotrophic levels (average heterotrophic community richness) considered in our study.”

The results show that the two versions of this index are very similar (Pearson’s $r = 0.98$), meaning that total community richness, as a measure of diversity across trophic levels, is not strongly influenced by a single trophic level (i.e. plants in this case). We provide an additional figure (Supplementary Figure 7c) and have added this information to the Results (L208-211):

“Excluding tree species richness and focusing only on heterotrophic community richness yielded congruent results (Supplementary Fig. 7c; Supplementary Table 9b) because both community richness metrics were highly correlated (Pearson’s $r = 0.98$, $P < 0.001$).”

As recommended by both Reviewers, we have replaced the term “multidiversity” by “average total community richness”, and we now also refer to species richness instead of diversity for the individual trophic levels.

Apparently there is no effect of succession and successional age on multifunctionality (see line 319-323 & Authors response to my queries about this). I am a bit surprised and find it hard to imagine that tree species of the young sites have similar characteristics as the tree species of the old sites? There are many examples in the ecological literature which show that early successional species are different from late successional species.

Following the logic of having no 'successional effect': Is it correct that in these forest no typical 'early successional' and 'late successional' species can be found (with early successional species, I mean species that colonize a disturbed habitat, that could even be a patch within a late successional forest). I would not be surprised if biological variables (which are a consequence of successional age) overrule and mask the importance of successional age.

-We agree that the general lack of effect of succession on multifunctionality is interesting. As you already mention, this might in part be explained by the fact that previous analyses at our study site only identified 3 out of 148 woody plant species to be specific to early or late successional stages (Bruehlheide et al. 2011 Ecol Monogr – cited in the MS). The community assembly of successional stages tended to follow neutral assembly rules, except that species accumulated over time. Moreover, previous analyses revealed that the functional diversity of the woody plant communities at our study site remained constant across the successional gradient (Böhnke et al. 2014 J Veg Sci – now cited in the revised MS). These findings might help explain why overall effects of stand age on multifunctionality were weak in our study. Moreover, we mention in the Discussion that while some diversity metrics were moderately (but not highly) correlated with stand age, the fact that biotic attributes were stronger predictors of multifunctionality than stand age suggests that changes in these biotic attributes are key to mechanistically explaining multifunctionality. We have expanded this part of the Discussion to include the additional information (L334-349):

“In part, the effects of trait diversity and species composition might be related to changes in successional and environmental conditions. The study plots were established along a successional gradient, and stand age showed significant (negative) effects on several of the individual functions considered in our study. Likewise, environmental conditions particularly influenced soil-related functions such as erosion control and microbial activity. Several predictors were moderately correlated with, and therefore potentially influenced by, stand age or environment (e.g. leaf chemical diversity as the variable most strongly correlated with stand age: Pearson’s $r = 0.61$, $P < 0.001$). However, the fact that the biotic attributes of the plant communities were stronger predictors of multifunctionality than abiotic variables, and that stand age or diversity interactions with stand age had only limited predictive power (as they were only retained in the final model of overall multidiversity but in none of the other models), suggests that changes in biotic attributes are key to mechanistically explaining multifunctionality. Previous studies at our study site found only few tree species that were specific to individual successional stages⁴⁸ and that woody plant functional diversity was maintained at a constant level throughout succession⁴⁹, which might contribute to explaining overall weak effects of stand age in our study.”

At the same time, we acknowledge that manipulative experiments are required to further disentangle the driving forces behind biodiversity–multifunctionality relationships (L349-351):

“We note the observational character of our study and that manipulative experiments are required to confirm causality and to further disentangle underlying mechanisms.”

It should be stated somewhere that this study describe a range of forest plots. Thus, the results presented are descriptive and in the end correlations among a wide range of variable are presented. Hence, it is important to state somewhere that such relationship are not based on causality and other underlying factors could be responsible for observed effects and relationships (e.g. is decomposer diversity really running the system, in terms of multifunctionality).

-We agree, and as already mentioned above, we have added a corresponding note to the Discussion (L349-351):

“We note the observational character of our study and that manipulative experiments are required to confirm causality and to further disentangle underlying mechanisms.”

On my request (see authors response), the authors now present residual scatterplots with data points. Please present the real data (no residual or transformed data). This will make it possible for readers and other researchers to compare values of this study with other studies (e.g. for instance, tree species richness of these forests or productivity or the number of decomposer species). Also, with residual data it is not possible (as reviewer) to verify whether the numbers reflect what one would typically expect for the studied forests.

-We agree that residual plots might not be ideal in that information on the change in absolute values of multifunctionality is not presented. We have revised all relevant figures (Figs 1, 3, 4, Supplementary Figs 3, 5, 8) to show observed data for multifunctionality and the raw data used in the statistical analyses for the x-axis (i.e. species richness, PCA axis score values, Rao's Q values; note that species richness for trophic levels with multiple taxa (e.g. herbivores, predators) necessarily represents the average of the scaled species richness across these taxa, as used in the statistical analyses). Moreover, we now provide an additional table (Supplementary Table 12) that shows details on the observed values for each individual ecosystem function variable and for each taxon considered in our study (mean, SD, minimum, maximum values per study plot). We hope that this helps to set the data into a broader context when comparing our results to other studies and systems.

We note, however, that showing observed values of multifunctionality does not account for the fact that diversity effects (and the corresponding regression lines) depend on the influence of the covariates included in the statistical models on multifunctionality (and therefore regression lines adjusted for covariates should be shown with data points adjusted for covariates, as they do not fit to raw data points and in extreme cases could even have an opposite direction). This is, for instance, illustrated by the effects of decomposer diversity, which depend on the structure of the woody plant communities (as also indicated by the SEM plot in Fig. 2). In addition to the figures using observed multifunctionality data, we therefore also provide the same figures as Supplementary Figures (Supplementary Figs 4, 6, 9, 11, 12), but with data points adjusted for covariates. We describe this procedure in the corresponding figure legends and also mention the adjustment on the y-axis labels of the figures. Because these figures represent the actual relationships between the predictors and multifunctionality after taking into account covariate influences, we would be in favor to show these figures in the main document instead of those based on observed data (which could then be transferred

to the Supplementary Material). However, we leave this decision to the Editor and Reviewer and will be fine if the current version with observed data (not corrected for covariates) in the main figures is preferred.

Line 178-179: I found this line unclear. What is meant exactly? Please add another sentence to clarify the goals.

-Revised for clarity as suggested. The sentence now reads (L184-188):

“We also found significant diversity effects on multifunctionality when analyzing these relationships on individual trophic levels in separate analyses, excluding functions directly mediated by the given trophic level being analyzed (e.g. parasitism for parasitoids, predation for predators). When considered separately, four of the six trophic levels showed significant diversity effects on multifunctionality ...”

Line 439: ITS58 does not exist. Please modify.

-Thanks for catching this typo (58 was the number of a citation that was not properly formatted). Changed to “ITS”.

Line 462: please rephrase sentence (is not completely clear): I assume x-min means the x-min of all studied plots?

- Rephrased for clarity (L489-491):

“Each of the 22 measurements of ecosystem functions were scaled to range from 0 to 1 with the formula $f(x) = (x_i - x_{min}) / (x_{max} - x_{min})$, where x is the variable of interest with its minimum (x_{min}) and maximum (x_{max}) values observed across all study plots...”

Reviewer #2 (Remarks to the Author):

The authors have done a commendable job in addressing my initial concerns. While I did not suggest it, I appreciate the application of SEM to look at the cascading effects, as it makes a lot of sense and I’m glad to see its inclusion.

- Thank you very much for helping to improve the previous manuscript with your suggestions!

I was overall happy with the revised manuscript, but I was still a little confused by the presentation. In Figure 1, only 4 aspects of diversity are considered (woody plant, chemical, decomposer, fungi), while in Figure 3, parasitoid, herbivore, and predator diversity are additionally considered (while woody plant diversity is not). The mismatch is not explained

in the text nor is it accounted for in the supplementary figures. I would instead present all aspects of diversity at once.

A more effective way to structure these results would be to have a single two-column figure, one with different aspects of diversity against multifunctionality (e.g., woody plant, etc.) and the second with the same diversity component against multifunctionality MINUS the functions related to that component. So, a combined figure 1 and 3. If this is unwieldy, the non-diversity components could be a different figure (e.g., PCA1 & 2, chemical diversity, etc.) since they are interpreted separately.

-We agree that the information provided in the text and figure legends might have been insufficient to clarify why these two figures were presented in this way. We have now clarified this in the Results (L144 and L184-188) as well as in the figure legends (Fig. 1, Fig. 3), where we explain that Fig. 1 presents the results of the overall model on multifunctionality, which only retained the strongest and most complementary predictors. In contrast, Fig. 3 presents the effects of diversity on multifunctionality when the analyses are restricted to individual trophic levels (as previously suggested by the Reviewer). We now also mention in the Results (and have added the information in new figures: Supplementary Figs 5 & 6) that multidiversity (now called total community richness) including all functions and multidiversity excluding functions specific to a given trophic level were highly correlated (now shown in the new Supplementary Table 8). This indicates that overall multidiversity was not strongly affected by the inclusion or exclusion of individual functions but rather represents a real average value across multiple functions. (L200-203):

“In general, the multifunctionality indices excluding individual functions were highly correlated with overall multifunctionality based on all functions (Supplementary Table 8), showing that none of the individual functions disproportionately influenced overall multifunctionality (Supplementary Figs 5 & 6).”

Because of these strong correlations, figures showing the results for both, overall multifunctionality and multifunctionality excluding individual functions, look nearly identical. We would therefore suggest showing the additional data on overall multifunctionality as a Supplementary Figure and have done so in our revision (Supplementary Figs 5 & 6). These can be compared to the figure that excludes individual functions (Fig. 3). Nevertheless, we also provide the figures as suggested by the Reviewer and attach them as an additional supplementary file to our revision (“Figure 3 alternate.pdf”). If this figure is preferred over the current Fig. 3, we will be happy to replace the current figure with this one.

Finally, a key finding from this paper is that diversity of various trophic levels is often ***negatively*** correlated with multifunctionality (Figures 1, 3). Possible reasons for this are given for fungi (lines 246-259), but I find it strange that, for example, herbivore, predator, and parasitoid diversity is always neutrally or positively correlated with individual functions (Figure 4). Yet, these predictors are generally neutrally or negatively correlated with average multifunctionality (Figure 3). Some explanation as to why this is would be warranted in the discussion.

-We now provide an explanation for this in the Discussion, using herbivore diversity as an example because this issue is most relevant for this trophic group (no discernible effect in Fig. 5, but negative effect in Fig. 3). The apparent mismatch is due to the fact that our

analyses only considered a maximum of four predictors (as suggested by the Reviewer in the previous round of reviews), meaning that only predictors with the strongest effects are retained in the reduced models. We have checked our results and confirmed that herbivore diversity tended to be negatively associated with several functions (decomposition, primary productivity, herbivory resistance), but other, stronger predictors masked this effect in the reduced models. However, the combined negative effects on several functions result in the negative effect of herbivore diversity on multifunctionality when analyses are restricted to specific trophic levels (as previously suggested by the Reviewer, and as presented in Figure 3). We have added this information to the Discussion (L264-269):

“Interestingly, the diversity of saprophytic fungi showed contrasting, negative effects on multifunctionality. The same was the case for herbivore diversity in the trophic level-specific analysis, although potentially negative effects on individual functions—such as primary productivity, herbivore resistance, or decomposition—were comparatively weak and masked by other, stronger diversity predictors in our analyses.”

Overall, nice effort, I think with cleaning up these minor issues, the paper should be very well received.

Minor comments:

Abstract: should include a mention of the (new tests of) cascading effects

-We have added this accordingly (L47-50):

“Moreover, cascading effects of higher trophic-level diversity on functions originating from lower trophic-level processes highlight that multitrophic biodiversity is key to understanding drivers of multifunctionality.”

Lines 79-83: this is a bit of a strawman argument. While its true that some studies consider diversity to be a ‘function’; many analyses are explicitly interested in it as a driver

-We adapted this part so that it is not mistaken as a claim that no study so far has focused on higher trophic level diversity as a driver of (multi)functionality (L82-86). Nevertheless, we think it is important to stress that a stronger focus on the effects of higher trophic levels is required in general.

“Perhaps even more important, however, is that the diversity of higher trophic-level organisms may serve as a key predictor of multifunctionality¹⁷, and therefore as a management target. Studies on multifunctionality sometimes consider heterotrophic diversity as one of the many components of multifunctionality^{10,18}, rather than as a direct driver of ecosystem functioning.”

Line 192: omit ‘multidiversity’; and call it ‘total community richness’;

- We have changed the term to “average total community richness” as suggested (adding “average” to indicate that standardized richness values were averaged across trophic levels).

Line 276-278: cite Supp Table 8 here, since this is not in the main figures

-Done (now Supplementary Table 10).

REVIEWERS' COMMENTS:

Reviewer #1 (Remarks to the Author):

The authors properly addressed the comments. In their response, the authors explain that they prefer to present plots using residual data for the main document and show the "real" data in the Supplement (now they did the other way around). I agree with the argumentation of the authors as long as the real data (Table 12 and various figures) are presented in the supplement.

Other comments: Macrofaunal decomposer species richness was positively related to multifunctionality (see line 257). This is interesting and relevant. However, I wonder whether the observed macrofaunal species richness reflects real values in the field (e.g. according to Table 12 species richness varied between 1 and 8 species per plot). I would imagine the macrofaunal species richness is an underestimate. Obviously, even if this is an underestimate, these values are relevant because I assume the same "underestimate" is made for each plot, making values comparable. Probably it is good to add a qualifier (e.g. in Table 12), and state that the observed values for some variables (e.g. macrofaunal decomposers, parasitoids, arbuscular mycorrhizal fungi, etc.) probably represent an underestimate. This because higher values of species richness for these groups have been reported in the literature (e.g. there was an underestimate due to the sampling time and frequency and because the molecular methods and primers used did not to assess the full microbial species richness).

REVIEWERS' COMMENTS:

Reviewer #1 (Remarks to the Author):

The authors properly addressed the comments. In their response, the authors explain that they prefer to present plots using residual data for the main document and show the *real* data in the Supplement (now they did the other way around). I agree with the argumentation of the authors as long as the real data (Table 12 and various figures) are presented in the supplement.

-Thank you for your helpful feedback! With the consent of Reviewer 1 we now present the observed data adjusted for covariates in the figures of the main document (Figs 1, 3, 4) and refer the reader to the corresponding figures showing raw data in the Supplementary Information (Figs 11, 12, 7a, Supplementary Table 12). Moreover, for all additional figures in the Supplementary Information, we provide both versions (data adjusted for covariates and raw data).

Other comments: Macrofaunal decomposer species richness was positively related to multifunctionality (see line 257). This is interesting and relevant. However, I wonder whether the observed macrofaunal species richness reflects real values in the field (e.g. according to Table 12 species richness varied between 1 and 8 species per plot). I would imagine the macrofaunal species richness is an underestimate. Obviously, even if this is an underestimate, these values are relevant because I assume the same *underestimate* is made for each plot, making values comparable. Probably it is good to add a qualifier (e.g. in Table 12), and state that the observed values for some variables (e.g. macrofaunal decomposers, parasitoids, arbuscular mycorrhizal fungi, etc.) probably represent an underestimate. This because higher values of species richness for these groups have been reported in the literature (e.g. there was an underestimate due to the sampling time and frequency and because the molecular methods and primers used did not to assess the full microbial species richness).

-As suggested, we have added a qualifier to Supplementary Table 12:

“Note that observed values are based on the taxa considered for a given functional group and might represent an underestimate of the total species richness of this group (e.g. macrofaunal decomposers, parasitoids, arbuscular mycorrhizal fungi) due to limitations of taxonomic scope, sampling time, or primers used in the molecular methods. Nevertheless, these limitations apply equally for all study plots, making data relevant and comparable among plots.”